# Inexact Alternating Direction Method of Multipliers with Efficient Local Termination Criterion for Cross-silo Federated Learning

## Abstract

Federated learning has attracted increasing attention in the machine learning community at the past five years. In this paper, we propose a new cross-silo federated learning algorithm with fast convergence guarantee to solve the machine learning models with nonsmooth regularizers. To solve this type of problems, we design an inexact federated alternating direction method of multipliers (ADMM). This method enables each agent to solve a strongly convex local problem. We introduce a new local termination criterion that can be quickly satisfied when using efficient solvers such as stochastic variance reduced gradient (SVRG). We prove that our method has faster convergence than existing methods. Moreover, we show that our proposed method has sequential convergence guarantees under the Kurdyka-Łojasiewicz (KL) assumption. We conduct experiments using both synthetic and real datasets to demonstrate the superiority of our new methods over existing algorithms.

## 1 Introduction

Federated learning (FL) is an emerging research paradigm in which multiple agents collaborate to solve a machine learning problem. Cross-silo FL is an important subclass where the participating agents are pre-defined silos, such as organizations or institutions (e.g., hospitals and banks) (Kairouz et al., 2021a). Typically, there are around 2-100 agents in this setting. Cross-silo federated learning finds significant applications in many domains such as medical and healthcare, finance, and manufacturing (Nandury et al., 2021; Huang et al., 2022; Yang et al., 2019). In a cross-silo federated learning (FL) task, each agent possesses a specific portion of the data, which they use to train their machine learning model locally. Once the local training is completed, all agents send their outputs to a central server. The server then aggregates these outputs and sends an update back to the participating agents.

Most FL works focus on the following federated composite optimization (Kairouz et al., 2021b; McMahan et al., 2017b; Pathak & Wainwright, 2020).

$$\min_{x \in \mathbb{R}^n} \sum_{i=1}^{p} f_i(x) + g(x), \tag{1}$$

where each $f_i : \mathbb{R}^n \to \mathbb{R}$ is smooth (probably nonconvex) and $L_i$-smooth, and $g : \mathbb{R}^n \to \mathbb{R} \cup \{+\infty\}$ is a proper closed convex regularizer. In machine learning applications, $f_i$ is the loss function of the agent $i$'s local data sets and $g$ can be $\ell_1$-regularizer, grouped $\ell_1$-regularizer, nuclear-norm regularizer (for matrix variable) (Candès & Recht, 2009; Bao et al., 2022), the indicator function of a convex constraint (Yuan et al., 2021; Bao et al., 2022), etc. Problem (1) is called the federated composite optimization in Yuan et al. (2021). In Yuan et al. (2021), the federated dual averaging (FedDualAvg) was proposed as an early attempt to deal with the nonsmooth $g$. Bao et al. (2022) proposed a fast federated dual averaging for problem (1) with a strongly convex $f$.

Although FedAvg, FedProx, FedDualAvg, and their variants have intuitive approaches to distribute tasks and aggregate local outputs, they face limitations in both theory and practice. For instance, Braverman et

Table 1: Comparison in the inner updates of federated splitting methods.

| Model | | Local Termination Criterion | Assumptions on $\epsilon_i^t$ | Local Solver | Local Complexity |
|---|---|---|---|---|---|
| $f_i$ | $g$ | | | | |
| FedSplit
Pathak & Wainwright (2020) | SC | 0 | $\|x_i^{t+1} - \mathrm{Prox}_{f_i}(\tilde{x}_i^t)\| \leq \epsilon_i^t$ | $\epsilon_i^t \leq \mathcal{O}(\epsilon)$ | GD | $\log(\epsilon^{-1})$ |
| FedPD
Zhang et al. (2021) | NC | 0 | $\mathbb{E}\|\nabla L_i(x_i^{t+1})\|^2 \leq \epsilon_i^t$ | $\epsilon_i^t \leq \mathcal{O}(\epsilon)$ | GD (SGD) | $\log(\epsilon^{-1})$ $(\epsilon^{-1})$ |
| FedDR
Tran-Dinh et al. (2021) | NC | NS | $\|x_i^{t+1} - \mathrm{Prox}_{f_i}(\tilde{x}_i^t)\| \leq \epsilon_i^t$ 
 $\|x_i^{t+1} - \mathrm{Prox}_{f_i}(\tilde{x}_i^t)\| \leq r\|x_i^{t+1} - x_i^t\|$ | $\frac{1}{p}\sum_{i=1}^p \sum_{t=0}^T \epsilon_i^t \leq \mathcal{O}(1)$ 
 None | - 
 - | - 
 - |
| FedADMM1
Gong et al. (2022) | NC | 0 | $\|\nabla L_i(x_i^{t+1})\|^2 \leq \epsilon_i^t$ | $\epsilon_i^t \leq \mathcal{O}(\epsilon)$ | - | - |
| FedADMM2
Zhou & Li (2022) | NC | 0 | $\|\nabla L_i(x_i^{t+1})\|^2 \leq \epsilon_i^t$ | $\epsilon_i^{t+1} \leq \nu_i \epsilon_i^t; \nu_i \in [1/2, 1)$ | - | $\log[(\epsilon_i^{t+1})^{-1}]$ |
| FedADMM3
Wang et al. (2022) | NC | NS | $\|x_i^{t+1} - \mathrm{Prox}_{f_i}(\tilde{x}_i^t)\| \leq \epsilon_i^t$ | $\frac{1}{p}\sum_{i=1}^p \sum_{t=0}^T \epsilon_i^t \leq \mathcal{O}(1)$ | - | - |
| FIAELT(Ours) | NC | NS | $\mathbb{E}_t^i\|x_i^{t+1} - \mathrm{Prox}_{f_i}(\tilde{x}_i^t)\|^2 \leq r_i\|x_i^t - \mathrm{Prox}_{f_i}(\tilde{x}_i^t)\|^2$ | **None** | SVRG | $\mathcal{O}(1)$ |

Table 2: Comparison in the server updates of the federated splitting methods in Table 1. SC = Strongly Convex, NC = Nonconvex, NS = Nonsmooth. $\epsilon$ is the same as in Table 1.

| Model | | Convergence | |
|---|---|---|---|
| $f_i$ | $g$ | Gradient | Sequence |
| FedSplit | SC | 0 | - | Linear |
| FedPD | NC | 0 | $O(T^{-1}) + \epsilon$ | - |
| FedDR | NC | NS | $O(T^{-1})$ | - |
| FedADMM1 | NC | 0 | $O(T^{-1}) + \epsilon$ | - |
| FedADMM2 | NC | 0 | $O(T^{-1})$ | - |
| FedADMM3 | NC | NS | $O(T^{-1})$ | - |
| FIAELT(Ours) | **NC** | NS | $O(T^{-1})$ | **Linear when $\theta \in (0, \frac{1}{2})$** |

al. McMahan et al. (2017a) demonstrated that FedAvg can diverge in certain scenarios. Even when FedAvg converges, as shown in Pathak & Wainwright (2020), the resulting fixed points may not necessarily be stationary points of the original problem. Additionally, the analysis in Yuan et al. (2021); Li et al. (2020a); Reddi et al. (2021) often assumes that the dissimilarity between agents is bounded, which may not hold in real-world applications. These shortcomings of existing methods motivate the exploration of federated splitting methods for solving (1). In general, the idea behind splitting methods in federated learning is to establish a connection between (1) and a constrained problem of the form:

$$\min_X \sum_{i=1}^p f_i(x_i) + g(x_1) \text{ s.t. } x_1 = x_2 = \cdots = x_p, \tag{2}$$

where $X = (x_1, x_2, \ldots, x_p)$.

Popular splitting methods in federated learning include FedSplit Pathak & Wainwright (2020), FedDR Tran-Dinh et al. (2021), FedPD Zhang et al. (2021), and ADMM based federated learning methods, Gong et al. (2022); Zhou & Li (2021); Zhang et al. (2021); Yue et al. (2021); Zhou & Li (2022). FedDR considers nonzero regularizer $g$ while FedSplit, FedPD, and FedADMM deal with the unregularized case where $g = 0$, which can not apply to the applications where regularizers are needed to induce sparse parameters Zou & Hastie (2005); Yuan et al. (2021) or low rank matrices Candès & Recht (2009); Bao et al. (2022).

At each round $t$ of federated splitting methods, each agent needs to find $x_i^{t+1}$ to approximate the proximal operator of each $f_i$ for the current point $\tilde{x}_i^t$ (denoted as $\mathrm{Prox}_{f_i}(\tilde{x}_i^t)$), via a number of local updates with a certain termination criterion. However, the number of local updates (defined as local complexity) required by existing criteria is either unexplored or tends to infinity with an infinitesimal tolerance $\epsilon$ as the number of server updates $T$ increases, as shown in Table 1. Therefore, a more advanced criterion that leads to a known constant number of more efficient local updates is much desired, which is an important goal of this work.

Moreover, existing federated splitting methods on nonconvex optimization with nonsmooth regularizer $g$ also only focus on the convergence rate of the gradient but ignore the convergence of the generated sequences to a desired critical point. Zhou & Li (2022); Yue et al. (2021) proves that the accumulation point is critical point

for regularized case ($g = 0$) but the convergence rate is still unknown. To obtain sequential convergence rate for nonsmooth regularizer $g \neq 0$ is also an important goal of this work.

## 1.1 Our Contributions

To fulfill the above two goals, we propose a novel splitting method called Federated Inexact ADMM with Efficient Local Termination (FIAELT) for the nonconvex nonsmooth composed optimization problem (1) in the context of cross-silo federated learning, based on the equivalence between (1) and an $np$-dimensional constrained problem (4). Compared with existing works on federated splitting methods, we summarize our contributions as follows.

• For the local update of our algorithm, we propose a new criterion $\mathbb{E}_t^i \|x_i^{t+1} - \text{Prox}_{f_i}(\tilde{x}_i^t)\|^2 \leq r_i \|x_i^t - \text{Prox}_{f_i}(\tilde{x}_i^t)\|^2$ (see Algorithm 1 for detail) where the tolerance $r_i \in (0, 1)$ does not need to be infinitesimal with large number $T$ of communication rounds. Hence, our local complexity can be $\mathcal{O}(1)$, which outperforms existing splitting methods with an unexplored or large number of local updates (see Table 1 for comparison). At the same time, we keep the state-of-the-art gradient convergence rate $\mathcal{O}(1/T)$ in the server updates (see Table 2).

• Furthermore, we demonstrate that FIAELT has sequential convergence properties in the deterministic case. Specifically, we prove that any accumulation point of the sequence generated at the server of FIAELT is a stationary point of (1). Moreover, we prove that FIAELT achieves global convergence under Kurdyka-Łojasiewicz (KL) geometry, which covers a wide range of functions in practice. Specifically, the server updates and the outputs of local servers converge in finitely many communications when the KL exponent $\alpha$ of the potential function is 0. These sequences converge linearly when $\alpha \in (0, \frac{1}{2})$. These sequences converge sublinearly when $\alpha \in (\frac{1}{2}, 1)$. In the analysis, our proposed new criterion plays a key role. To the best of our knowledge, FIAELT is the first federated learning method that has sequential convergence rate in nonconvex nonsmooth settings.

• Finally, we conducted experiments involving the training of fully-connected neural networks. In these experiments, we compared our method against existing splitting methods as well as other state-of-the-art Federated methods. The experimental results revealed that our method is competitive and consistently outperforms the other approaches in terms of training loss, training accuracy, and testing accuracy. These findings indicate the superior performance and effectiveness of our proposed method in the task of training fully-connected neural networks.

## 1.2 Related Work

The literature of federated learning is rich. In this work, we only focus on the splitting methods in federated learning. A comparison between our method and existing splitting methods is summarized in Table 1.

In Pathak & Wainwright (2020), FedSplit was proposed. It implements the Peaceman-Rachford splitting method for (2). Pathak & Wainwright (2020) analyzed the proposed method in the case where $g = 0$ and $\sum_i f_i$ is strongly convex. Pathak & Wainwright (2020) showed that when the error between the local output and the $\text{Prox} f_i$ is under a threshold $\epsilon$, the sequence generated at the server by FedSplit linearly converges to an inexact solution of (1) up to an error determined by $\epsilon$. They also applied the FedSplit to a strongly convex majorization of the original problem. In this setting, they showed a complexity of $\tilde{O}(\sqrt{\epsilon})$ to obtain an $\epsilon$-optimal function value. However, in general convex settings, it assumes FedSplit locally computes $\text{Prox} f_i$ exactly, which is unrealistic when the local server solves large-scale problems.

When $g = 0$, there are several work on federated ADMM, Zhang et al. (2021); Gong et al. (2022); Zhou & Li (2022); Elgabli et al. (2022). Gong et al. (2022) proposed FedADMM that randomly selects agents to attend each round. The $i_{\text{th}}$ agent terminates the local iterations when the norm of the local gradient of the current iterate is under a threshold $\epsilon_i$. When there is an upper bound $\epsilon$ for $\{\epsilon_i\}$, they showed FedADMM has a complexity of $O(\epsilon^{-1}) + O(\epsilon)$ to reach an $\epsilon$-surrogate stationary point. When $f_i$'s are twice differentiable, ADMM is applied in designing a second-order FL method in Elgabli et al. (2022). Zhou & Li (2022) proposed an inexact ADMM for federated learning problems. At round $t$, the $i_{\text{th}}$ agent terminates the local updates when the norm of the local gradient is under a threshold $\epsilon_i^t$. They assume $\{\epsilon_i^t\}_t$ decreases exponentially, i.e.,

$\epsilon_i^{t+1} \leq \nu_i \epsilon_i^t$ with $\nu_i \in [\frac{1}{2}, 1)$. They showed that the generated sequence accumulates at the stationary point. By further assuming the accumulation point of the generated sequence is isolated, they show the generated sequence converges globally. Compared with this work, we do not assume the accumulation point of the generated sequence to be isolated when we analyze the sequential convergence of our method.

When $g \neq 0$, Tran-Dinh et al. (2021) proposed FedDR that applies the Douglas-Rachford (DR) splitting algorithms for (2). They combined the DR method with randomized block-coordinate strategies and asynchronous implementation. They estimated the complexity of FedDR under different termination criteria for local updates. The termination criteria in Tran-Dinh et al. (2021) test whether the distance between the prox of $f$ and its approximation can be bounded by a certain value. However, this distance is unable to check in practice, especially when we use stochastic gradient methods for local updates. Yue et al. (2021) also considered the case where $g \neq 0$. Specifically, they considered the case when $g$ is the Bregman distance. Assuming the Hessian of $f_i$'s in (1) being Lipschitz continuous, Yue et al. (2021) showed any accumulation point of the generated sequence is a stationary point. Yue et al. (2021) also showed the proposed method has a complexity of $O(\epsilon^{-1})$ to reach an $\epsilon$-stationary point.

## 2 Preliminaries

In this paper, we denote $\mathbb{R}^n$ the $n$-dimensional Euclidean space with inner product $\langle \cdot, \cdot \rangle$ and Euclidean norm $\|\cdot\|$. We denote the set of all positive numbers as $\mathbb{R}_{++}$. We denote the distance from a point $a$ to a set $\mathcal{A}$ as $d(a, \mathcal{A})$. For a random variable $\xi$ defined on a probability space $(\Xi, \Sigma, P)$, we denote its expectation as $\mathbb{E}\xi$. Given an event $A$, the conditional expectation of $\xi$ is denoted as $\mathbb{E}(\xi|A)$.

An extended-real-valued function $f : \mathbb{R}^n \to [-\infty, \infty]$ is said to be proper if $\text{dom} f = \{x \in \mathbb{R}^n : f(x) < \infty\}$ is not empty and $f$ never equals $-\infty$. We say a proper function $f$ is closed if it is lower semicontinuous. We define the indicator function of a closed set $\mathcal{A}$ as $\delta_{\mathcal{A}}(x)$, which is zero when $x \in \mathcal{A}$ and $\infty$ otherwise.

We define the regular subdifferential of a proper function $f : \mathbb{R}^n \to [-\infty, \infty]$ at $x \in \text{dom} f$ as $\hat{\partial} f(x) := \left\{ \xi \in \mathbb{R}^n : \liminf_{z \to x, \ z \neq x} \frac{f(z) - f(x) - \langle \xi, z - x \rangle}{\|z - x\|} \geq 0 \right\}$ The (limiting) subdifferential of $f$ at $x \in \text{dom} f$ is defined as $\partial f(x) := \left\{ \xi \in \mathbb{R}^n : \exists x^k \xrightarrow{f} x, \xi^k \to \xi \text{ with } \xi^k \in \hat{\partial} f(x^k), \forall k \right\}$, where $x^k \xrightarrow{f} x$ means both $x^k \to x$ and $f(x^k) \to f(x)$. For $x \notin \text{dom} f$, we define $\hat{\partial} f(x) = \partial f(x) = \emptyset$. We denote $\text{dom} \partial f := \{x : \partial f(x) \neq \emptyset\}$. For a differential function $h : \mathbb{R}^m \times \mathbb{R}^n \to \mathbb{R}^l$, we denote $\nabla_x L(x, y)$ and $\nabla_y L(x, y)$ as the partial derivatives with respect to $x$ and $y$ correspondingly. We defined the normal cone of a set $\mathcal{A}$ at $x$ as $N_{\mathcal{A}}(x) := \partial \delta_{\mathcal{A}}(x)$. For a proper function $f : \mathbb{R}^n \to [-\infty, \infty]$, we denote the proximal operator of $f$ as $\text{Prox}_{\alpha f}(x) = \text{Arg} \min_{z \in \mathbb{R}^n} \left\{ f(z) + \frac{1}{2\alpha} \|z - x\|^2 \right\}$. Consider a problem $\min f + g$, where $f$ is a smooth function and $g$ is properly closed convex. We say $x$ is a stationary point of this problem when $0 \in \nabla f(x) + \partial g(x)$. We say $x$ is an $\varepsilon$-stationary point if $d^2(0, \nabla f(x) + \partial g(x)) \leq \varepsilon$.

We next introduce the KL property used in analyzing the sequential convergence. Let $\Psi_a$ be defined as the set of concave functions $\psi : [0, a) \to [0, \infty)$ satisfying $\psi(0) = 0$, being continuously differentiable on $(0, a)$, and satisfying $\psi' > 0$ on $(0, a)$.

**Definition 1 (Kurdyka-Łojasiewicz property and exponent).** *A proper closed function $f : \mathbb{R}^n \to (-\infty, \infty]$ is said to satisfy the Kurdyka-Łojasiewicz (KL) property at an $\hat{x} \in \text{dom} \partial f$ if there are $a \in (0, \infty]$, a neighborhood $V$ of $\hat{x}$ and a $\varphi \in \Psi_a$ such that for any $x \in V$ with $f(\hat{x}) < f(x) < f(\hat{x}) + a$, it holds that $\psi'(f(x) - f(\hat{x}))\text{dist}(0, \partial f(x)) \geq 1$. If $f$ satisfies the KL property at $\hat{x} \in \text{dom} \partial f$ and $\psi$ can be chosen as $\psi(\nu) = a_0 \nu^{1-\alpha}$ for some $a_0 > 0$ and $\alpha \in [0, 1)$, then we say that $f$ satisfies the KL property at $\hat{x}$ with exponent $\alpha$. A proper closed function $f$ satisfying the KL property with exponent $\alpha \in [0, 1)$ at every point in $\text{dom} \partial f$ is called a KL function with exponent $\alpha$.*

Functions satisfying KL property includes proper closed semi-algebraic functions, the quadratic loss function plus possibly nonconvex piecewise linear regularizers Attouch et al. (2010); Li & Pong (2018); Attouch et al. (2013); Zeng et al. (2021).

## 3   Federated Inexact ADMM with Efficient Termination Criterion

We relate the problem (1) to (2). For (2), we view it as the following $np$-dimensional problem:

$$\min_{X \in \mathbb{R}^{np}} F(X) + G(X), \tag{3}$$

where $X = (x_1, x_2, \ldots, x_p)$ with each $x_i \in \mathbb{R}^n$, $F(X) := \sum_{i=1}^p f_i(x_i)$ with $f_i$'s in (1), $G(X) := g(x_1) + \delta_{\mathfrak{C}}(X)$ with $\mathfrak{C} := \{X : x_1 = \cdots = x_p\}$ and $g$ in (1).

The following proposition establishes the relation between (3) and (1).

**Proposition 1.** *If $X^* = (x_1^*, \ldots, x_p^*)$ is a stationary point of* (3)*, then $x_1^*$ is a stationary point of* (1)*. Furthermore, if $X = (x_1, \ldots, x_p)$ is an $\varepsilon$-stationary point of* (1)*, then $x_1$ is a $p\varepsilon$-stationary point of* (1)*.*

Based on this relation, we consider ADMM to solve (3). Rewrite (3) as the following equivalent problem:

$$\min_{X,Y \in \mathbb{R}^{np}} F(X) + G(Y) \text{ s. t. } X = Y. \tag{4}$$

The augmented lagrangian function of (4) is defined as:

$$L_\beta(X, Y, Z) := F(X) + G(Y) + \langle X - Y, Z \rangle + \frac{\beta}{2} \|X - Y\|^2. \tag{5}$$

Given a starting point $(X^0, Y^0, Z^0) \in \mathbb{R}^{np} \times \mathbb{R}^{np} \times \mathbb{R}^{np}$ and $\tau, \beta > 0$, the ADMM for (3) is as follows:

$$\begin{cases} X^{t+1} = \arg\min_X L_\beta(X, Y^t, Z^t), \\ Z^{t+1} = Z^t + \tau\beta(X^{t+1} - Y^t), \\ Y^{t+1} = \arg\min_Y L_\beta(X^{t+1}, Y, Z^{t+1}). \end{cases} \tag{6}$$

Now we give an equivalent form of the third equation in (6) as follows.

**Proposition 2.** *Consider* (3)*. Let $\{(X^{t+1}, Y^{t+1}, Z^{t+1})\}$ be generated by* (6)*. Suppose $\beta > \max_i L_i$. Then the solution of the problem in the third equation of* (6) *is $(y_1, \ldots, y_1)$ with $y_1 = \mathrm{Prox}_{\frac{1}{\beta p}g}(\frac{1}{p}\sum_{i=1}^p (x_i^{t+1} + \frac{1}{\beta}z_i^{t+1})))$.*

On the other hand, since $F(X)$ in (3) is separable, we can write $L_\beta(X, Y, Z)$ in (5) as $L_\beta(X, Y, Z) = \sum_{i=1}^p L_{\beta,i}(x_i, y_i, z_i)$, where

$$L_{\beta,i}(x_i, y_i, z_i) := f_i(x_i) + \langle x_i - y_i, z_i \rangle + \frac{\beta}{2}\|x_i - y_i\|^2.$$

Therefore, the first equality in (6) can be rewritten as $x_i^{t+1} = x_{i,*}^{t+1}$ where

$$x_{i,*}^{t+1} := \underset{x_i}{\arg\min} L_{\beta,i}(x_i, y^t, z_i^t); i = 1, \ldots, p. \tag{7}$$

In practice, (7) cannot be exactly solved as $f_i$ is usually a nonconvex loss function involving large training data. Hence, existing federated splitting methods inexactly solve (7) up to a certain local criterion. However, the computational complexities of the local updates required by these criteria are either unexplored or very large (see Table 1). To solve this limitation, we propose the following criterion.

$$\mathbb{E}_t^i \|x_i^{t+1} - x_{i,\star}^{t+1}\|^2 \leq r_i \|x_i^t - x_{i,\star}^{t+1}\|^2. \tag{9}$$

where $\mathbb{E}_t^i$ denotes conditional expectation given the past trajectory $\{(x_i^s, y^s, z_i^s) : s = 0, 1, \ldots, t\}$, and the tolerance $r_i \in (0, 1)$ does not need to be arbitrarily small to ensure $\mathcal{O}(1)$ local complexity even with stochastic gradient, as will be shown in the convergence analysis.

---

**Algorithm 1** Federated Inexact ADMM with Efficient Local Termination (FIAELT) for (1)

---
1: **Input:** $\beta, \tau > 0$, $r_i > 0$, $m_i \in \mathbb{N}_+$, $\eta_i > 0$. $(x_i^0, y_i^0, z_i^0)$ and $\bar{x}^0 = \frac{1}{p}\sum_i x_i^0$, $\bar{z}^0 = \frac{1}{p}\sum_i z_i^0$ for agents $i = 1, \ldots, p$.
2: **for** iteration $t = 0, 1, \ldots, T-1$ **do**
3:      **for** agent $i = 1, \ldots, p$ in parallel **do**
4:          Find $x_i^{t+1}$ to approximately solve:

$$x_i^{t+1} \approx \min_{x_i} L_{\beta,i}(x_i, y_i^t, z_i^t) := x_{i,\star}^{t+1}. \qquad (8)$$

     such that the criterion (9) is satisfied.
         Upload $\Delta_{x_i,t+1} = x_i^{t+1} - x_i^t$ and $\Delta_{z_i,t+1} = \tau\beta(x_i^{t+1} - y_i^t)$ to the server.
5:      **end for**
6:      The server calculates $\bar{x}^{t+1} = \bar{x}^t + \frac{1}{p}\sum_i \Delta_{x_i,t+1}$, $\bar{z}^{t+1} = \bar{z}^t + \frac{1}{p}\sum_{i=1}^p \Delta_{z_i,t+1}$ and $y^{t+1} = \text{Prox}_{\frac{1}{\beta p}g}(\bar{x}^{t+1} + \frac{1}{\beta}\bar{z}^{t+1})$, and broadcasts these variables to each agent.
7: **end for**

---

We propose Algorithm 1 that implements the ADMM rule (6) in a federated way, where $x_i^{t+1}$ inexactly solves (7) with stochastic gradient methods.

When $\beta > L := \max_i L_i$, the local problem (8) is minimizing a strongly convex smooth function that has Lipscihtz continuous gradient. Hence, using the stochastic method called SVRG in Johnson & Zhang (2013), we obtain $x^{t+1}$ that satisfies the following property.

**Proposition 3.** *Consider* (1). *Set* $\beta > L := \max_i L_i$. *Let* $\{(x_i^t, y_i^t, z_i^t)\}$ *be generated by Algorithm 1. Using SVRG in Johnson & Zhang (2013) with Option II with frequency* $m_i$, *learning rate* $\eta_i$, *and initialization* $x_i^t$ *for* (8), *such that*

$$\frac{1}{\eta_i(\beta - L_i)(1 - 2\eta_i(\beta + L_i))m_i} + \frac{2\eta_i(\beta + L_i)}{1 - 2\eta_i(\beta + L_i)} =: \rho_i < 1. \qquad (10)$$

*Then criterion* (9) *is satisfied in at most* $k_t^i = \log_{1/\rho_i} \frac{\beta + L_i}{r_i(\beta - L_i)}$ *iterations of SVRG.*

**Remark 1.** *The above proposition shows that fixing any* $r_i \in (0, 1)$, *SVRG outputs an inexact solution of the local subproblem* (8) *within* $\mathcal{O}(1)$ *steps, independent of the number of communication rounds* $T$. *In contrast, the number of local updates required by other existing federated splitting methods is either unexplored or increases to infinity with* $T$.

**Remark 2.** *When* (9) *is deterministic, our subproblem degenerates to minimizing a strongly convex function. According to the well know results, minimizing a strongly convex function with the simplest gradient descent method produce a linear convergent sequence of variables. Following the same analysis in the proofs of Proposition 3, we will have the local complexity of order* $O(1)$.

## 4 Convergence Analysis of Algorithm 1

We analyze the convergence properties of the variables $X^t := [x_1^t; \ldots; x_p^t]$, $Y^t := [y_1^t; \ldots; y_p^t]$, $Z^t := [z_1^t; \ldots; z_p^t]$ generated by Algorithm 1. We also denote $L := \max_i L_i$, $r := \max_i r_i$, $X_*^{t+1} := [x_{1,*}^{t+1}; \ldots; x_{p,*}^{t+1}]$ and $W = \inf_X F(X) + \inf_Y G(Y) > -\infty$ throughout the paper. First, the update rules of Algorithm 1 can be rewritten into the combined vectors $X^t, Y^t, Z^t$ as follows.

We first show the following property.

**Proposition 4.** *The update rules in Algorithm 1 satisfy*

$$\mathbb{E}\|X^{t+1} - X_\star^{t+1}\|^2 \le r\|X^t - X_\star^{t+1}\|^2, \qquad (11)$$

$$Z^{t+1} = Z^t + \tau\beta(X^{t+1} - Y^t), \qquad (12)$$

$$Y^{t+1} = \min_Y L_\beta(X^{t+1}, Y, Z^{t+1}), \qquad (13)$$

With Proposition 4, we can analyze $\{(X^t, Y^t, Z^t)\}$ to analyze the convergence properties of Algorithm 1. For $\{(X^t, Y^t, Z^t)\}$, we have the following theorem that is important in establishing our main convergence properties.

**Proposition 5.** *Select hyperparameters* $\beta \geq 5L$, $r_i \in (0, 0.01]$, $\tau \in [1/2, 1)$. *Denote* $\Gamma := \frac{1-\tau}{\tau}$, $\Theta = 2\beta^2 + 4L^2$, $\Lambda := 4L^2$. $\Upsilon := \frac{\Theta}{\tau\beta}\frac{4r}{1-2r}$ *and* $\delta := \frac{1}{4}(\beta - L) - 2\Upsilon$. *Define*

$$H(X, Y, Z, X', Z') := L_\beta(X, Y, Z) + \frac{\Gamma}{\tau\beta}\|Z - Z'\|^2 + \Upsilon\|X - X'\|^2.$$

*and* $H_{t+1} := \mathbb{E}H(X^{t+1}, Y^{t+1}, Z^{t+1}, X^t, Z^t)$. *Then for* $t \geq 1$, *it holds that* $\delta \geq 0.1L$ *and*

$$H_{t+1} \leq H_t - \delta\mathbb{E}\|X^{t+1} - X^t\|^2 - \frac{\beta}{2}\mathbb{E}\|Y^{t+1} - Y^t\|^2. \tag{14}$$

*Hence, the sequence* $\{H_t\}$ *converges to some* $H_* \geq W$.

Thanks to Proposition 5, we have the following property with respect to the successive changes.

**Corollary 1.** *Consider* (1) *and let* $(X^t, Y^t, Z^t)$ *be defined as in Proposition 4. Suppose assumptions in Proposition 5 hold. Then* $\lim_t \mathbb{E}\|X^t - X^{t+1}\|^2 = \lim_t \mathbb{E}\|Y^{t+1} - Y^t\|^2 = \lim_t \mathbb{E}\|Z^{t+1} - Z^t\|^2 = \lim \mathbb{E}\|Y^t - X^t\|^2 = 0$.

**Remark 3.** *Corollary 1 together with Propositions 1 and 4 shows that the expectations of successive changes of* $\{(x_1^t, \ldots, x_p^t, y^t, z_1, \ldots, z_p^t)\}$ *generated by Algorithm 1 also converge to 0.*

Based on Proposition 5, $\{(X^t, Y^t, Z^t)\}$ has the following convergence property.

**Theorem 1.** *Select hyper-parameters per Proposition 5 hold and let* $H_*$ *be defined as in Proposition 5. Then*

$$\sum_{t=0}^{T} \mathbb{E}\|\nabla F(Y^{t+1}) + \xi^{t+1}\|^2 \leq D\left(\|\nabla L_\beta(X^0, Y^0, Z^0)\|^2 + \|X^0 - Y^0\|^2\right) + D\left(L_\beta(X^0, Y^0, Z^0) - W\right), \tag{15}$$

*where*

$$D := \max\{3(L+\beta)^2 \frac{2r}{1-2r}, \left(\frac{L}{\tau\beta} + 1\right)^2, (L+\beta)^2\} \cdot \max\{D_1, D_2, D_3\} \tag{16}$$

*with* $D_1 := \frac{2\Gamma + \Theta\frac{8r}{1-2r} + 2}{\min\{\delta, \frac{1}{2}\beta\}}$, $D_2 := (1 + \Gamma)\frac{3(r+1)}{(L-\beta)^2} + D_1\frac{4}{(L-\beta)^2}\left(\frac{L+\beta+1}{2} + 2\tau\beta(\Gamma + 1) + \Upsilon + \frac{(L-\beta)^2}{8}\right)$, $D_3 := \max\{3, D_1 2\tau\beta(\Gamma + 1)\}$, $\Gamma$, $\Upsilon$ *and* $\Theta$ *being defined in Proposition 5.*

Combining Theorem 1 with Proposition 1 and Proposition 3, we immediately obtain the following convergence rate of Algorithm 1.

**Corollary 2.** *Select hyperparameters* $\beta = 5L$, $r_i = 0.005$, $\tau = 1/2$ *in Algorithm 1. Then the following convergence rate holds.*

$$\frac{1}{1+T}\sum_{t=0}^{T} \mathbb{E}d^2(0, \sum_i \nabla f_i(y^{t+1}) + \partial g(y^{t+1}))$$
$$\leq pD\left(\|\nabla L_\beta(X^0, Y^0, Z^0)\|^2 + \|X^0 - Y^0\|^2\right) + pD\left(L_\beta(X^0, Y^0, Z^0) - W\right).$$

*where* $D$ *is the one defined in Theorem 1. Furthermore, the criterion* (9) *can be satisfied by implementing 10 iterations of SVRG Johnson & Zhang (2013) with Option II with frequency* $m_i = 200$, *learning rate* $\eta_i = \frac{1}{40L}$, *and initialization* $x_i^t$ *for* (8).

**Remark 4.** *Corollary 2 indicates that compared with existing federated methods, we keep the same state-of-the-art convergence rate* $\mathcal{O}(1/T)$ *with* $T$ *being the number of the communication round, while only* $\mathcal{O}(1)$ *local update steps for the local* (8) *is required.*

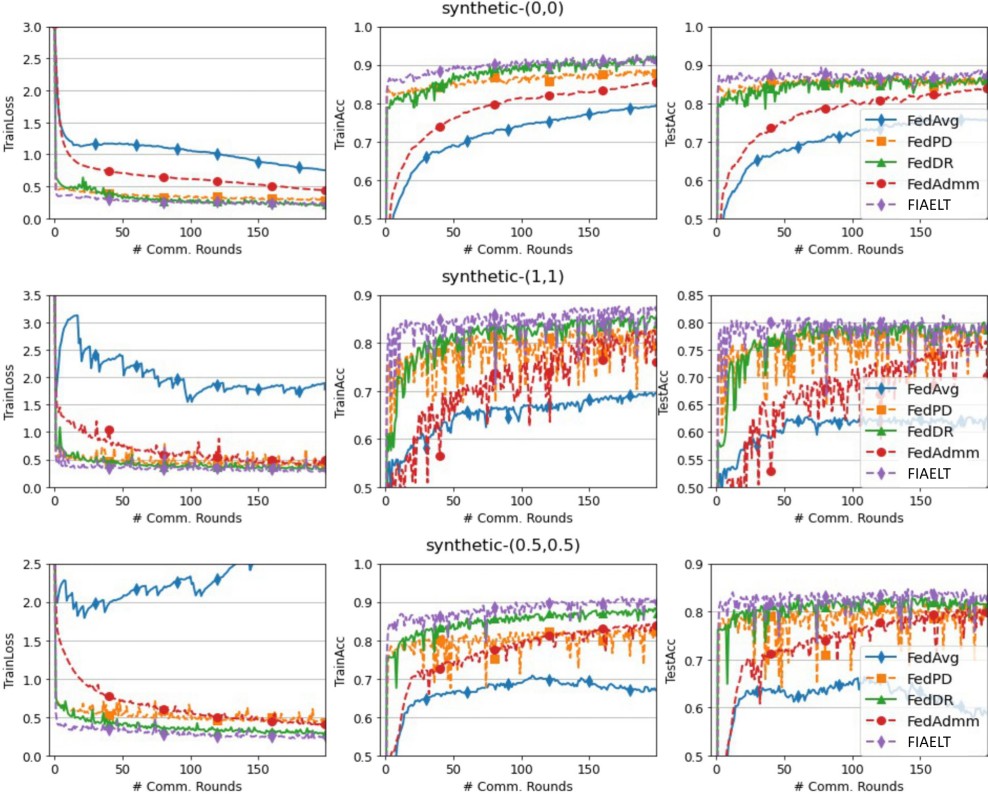

Figure 1: Results on Synthetic-$\{(0,0), (0.5, 0.5), (1,1)\}$ dataset.

## 4.1 Sequential Convergence in the Deterministic Case

In this section, we further investigate the convergence of the sequence $\{(X^t, Y^t, Z^t)\}$ generated by Algorithm 1 when (9) holds deterministically, i.e., holds without the expectation. We first show the properties of the set of accumulation points of $\{(X^t, Y^t, Z^t, X^{t-1}, Z^{t-1})\}$.

**Proposition 6.** *Consider* (1) *and let* $\{(X^t, Y^t, Z^t)\}$ *be generated by Algorithm 1 with* (9) *holding deterministically. Suppose assumptions in Proposition 5 hold. Suppose* $\{(X^t, Y^t, Z^t)\}$ *is bounded. Then any accumulation point of* $\{Y^t\}$ *is a stationary point of* (3).

Combining Proposition 6 with Proposition 1 and Proposition 2, we immediately have the subsequential convergence of the sequence generated by FIAELT.

**Corollary 3.** *Let* $\{(x_1^t, \ldots, x_p^t, y^t, z_1^t, \ldots, z_p^t)\}$ *be generated by Algorithm 1 with* (9) *holding deterministically. Let* $(X^t, Y^t, Z^t)$ *be defined as in Proposition 4. Suppose assumptions in Proposition 6 hold. Then any accumulation point of* $\{y^t\}$ *is a stationary point of* (1).

Next, we present the convergence rate of $(X^t, Y^t, Z^t)$.

**Theorem 2.** *Consider* (1) *and Algorithm 1 with* (9) *holding deterministically. Let* $(X^t, Y^t, Z^t)$ *be defined as in Proposition 4. Suppose assumptions in Proposition 5 hold. Let* $H$ *be defined as in Proposition 5 and suppose* $H$ *is a KL function with exponent* $\alpha \in [0,1)$. *Then* $\{(X^t, Y^t, Z^t)\}$ *converges globally. Denoting* $(X^*, Y^*, Z^*) := \lim_t (X^t, Y^t, Z^t)$ *and* $d_s^t := \|(X^t, Y^t, Z^t) - (X^*, Y^*, Z^*)\|$, *then the followings hold. If* $\alpha = 0$, *then* $\{d_s^t\}$ *converges finitely. If* $\alpha \in (0, \frac{1}{2}]$, *then there exist* $b > 0$, $t_1 \in \mathbb{N}$ *and* $\rho_1 \in (0,1)$ *such that* $d_s^t \le b\rho_1^t$ *for* $t \ge t_1$. *If* $\alpha \in (\frac{1}{2}, 1)$, *then there exist* $t_2$ *and* $c > 0$ *such that* $d_s^t \le ct^{-\frac{1}{4\alpha - 2}}$ *for* $t \ge t_2$.

**Remark 5.** *Proposition 3 and Theorem 2 jointly illustrate that the local outputs* $\{x_i^t\}_t$ *and the server updates* $y^t$ *achieve global linear convergence towards a stationary point of* (1) *when the Kurdyka-Lojasiewicz (KL) exponent of function* $H$ *is set to* $\frac{1}{2}$. *The precise determination of the KL exponent of* $H$ *is interconnected*

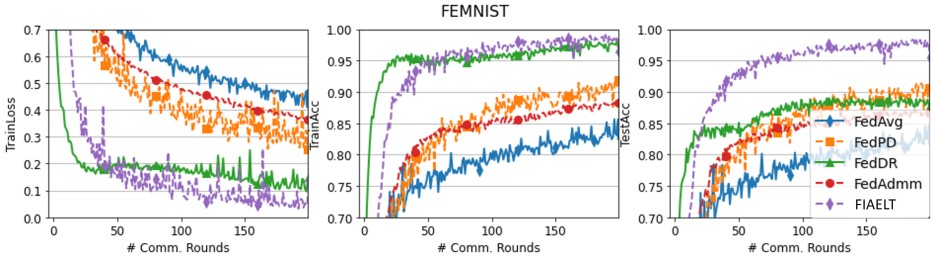

Figure 2: Results on FEMNIST dataset.

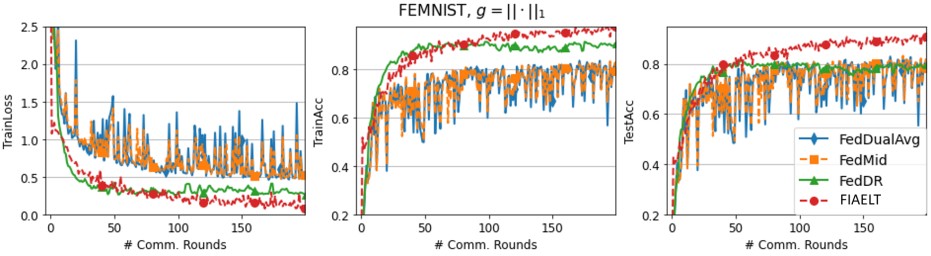

Figure 3: Results on FEMNIST dataset with $\mathbb{L}_1$-norm.

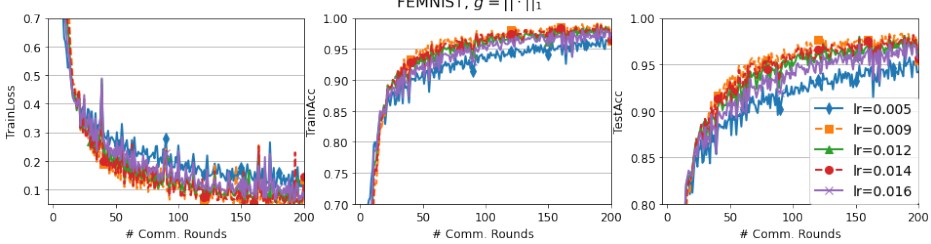

Figure 4: Results of our algorithm on FEMNIST dataset with different learning rates. ($\mathbb{L}_1$-norm regularize.)

*with another aspect involving the investigation of error bounds, which is beyond the boundaries of the present paper's scope. Interested readers are referred to sources such as Attouch et al. (2010); Li & Pong (2018); Attouch et al. (2013); Zeng et al. (2021) for more deeper insights.*

## 5 Experimental Results

To evaluate the performance of our proposed FIAELT algorithm, we conduct experiments on both realistic and synthetic datasets. When $g = 0$ in (1), we compare our algorithm with FedDRTran-Dinh et al. (2021), FedPD Zhang et al. (2021), FedAvg McMahan et al. (2017b), FedAdmm Zhou & Li (2022). When $g = \lambda\|\cdot\|_1$ for some $\lambda \in \mathbb{R}_{++}$, we compare our algorithm with FedMid Yuan et al. (2021), FedDualAvg Yuan et al. (2021), and FedDR. Following FedDR Tran-Dinh et al. (2021), we choose the neural network as our model, and the details are deferred to the supplementary materials. For FedDR, FedPD, we refer to the code provided in Tran-Dinh et al. (2021), and we also re-implement the FedAdmm based on them. All experiments are running on the Linux-based server with the configuration: 8xA6000 GPU with 48GB memory each. To be in accordance with the theoretical analysis, we sample all the clients to perform updates for our algorithm in each communication round. We pick up hyper-parameters carefully and show the best results for each algorithm. For evaluation metrics, we use training loss, training accuracy, and test accuracy. Our code is available at https://anonymous.4open.science/r/FIAELT_TMLR-D6C7/.

**Results on synthetic datasets.** Following the data generation process on Li et al. (2020a); Tran-Dinh et al. (2021), we generate three datasets: `synthetic-{(0,0), (0.5, 0.5), (1,1)}`. All agents perform

updates at each communication round. Our algorithm is compared using synthetic datasets in both iid and non-iid settings. The performance of five algorithms on non-iid synthetic datasets is shown as Figure 1. Our algorithm can achieve better results than FedPD, FedAdmm, FedAvg, and FedDR on all three synthetic datasets.

FEMNIST Cohen et al. (2017); Caldas et al. (2018) dataset is a more complex and federated extended MNIST. It has 62-class (26 upper-case and 26 lower-case letters, 10 digits) and the data is distributed to 200 devices. Figure 2 depicts the results of all 5 algorithms on FEMNIST. As it shows, FIAELT can achieve comparable training accuracy and loss value with FedDR. In comparison with FedAdmm, FedPD, and FedAvg, FIAELT has a significant improvement in both training accuracy and loss value. Our algorithm can also work much better with test accuracy than the other 4 algorithms.

**Results with the $\mathbb{L}_1$ norm.** Following FedDR Tran-Dinh et al. (2021), we also consider the composite setting with $g(x) := 0.01\|x\|_1$ to verify our algorithm by selecting different learning rates and the number of local SGD epochs. We conduct the experiment on the FEMNIST dataset and we show the results as Figure 3. As we can see from the training loss and training accuracy, FIAELT has competitive efficiency with FedDR and outperform FedDualAvg and FedMid. In addition, in testing accuracy, FIAELT outperforms all the other methods. Figure 5 shows how different learning rates affect the performance of our FIAME on the FEMNIST dataset.

## 6 Conclusion

In this paper, we propose a federated inexact ADMM with a new local termination criterion. This criterion is efficient and can be satisfied within iterations unrelated to the communication rounds, particularly when using stochastic gradient methods as the local solver. Our new method has the best-known complexity while having efficient local updates. Additionally, we provide proof that the proposed method has sequential convergence guarantees in the deterministic case. Under KL assumptions, we demonstrate that the whole generated sequence can converge sublinearly, linearly, or even finitely. Our experiments consistently demonstrate that the proposed method consistently outperforms state-of-the-art methods, especially in terms of testing accuracy.

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

| Dataset | Size(Input x FC layer x Output) |
|---------|-------------------------------|
| Synthetic | 60 x 32 x 10 |
| MNIST | 784 x 128 x 10 |
| FEMNIST | 784 x 128 x 26 |

Table 3: The details of the neural networks in our numerical experiments.

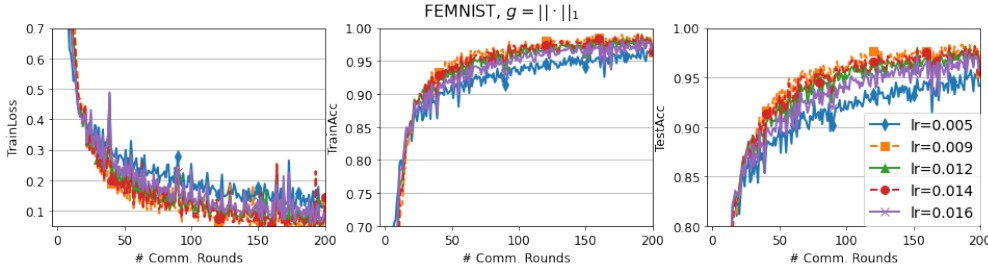

Figure 5: Results of our algorithm on FEMNIST dataset with different learning rates. ($\mathbb{L}_1$-norm regularize.)

Honglin Yuan, Manzil Zaheer, and Sashank J. Reddi. Federated composite optimization. In *Proceedings of the 38th International Conference on Machine Learning, ICML 2021, 18-24 July*, 2021.

Sheng Yue, Ju Ren, Jiang Xin, Sen Lin, and Junshan Zhang. Inexact-admm based federated meta-learning for fast and continual edge learning. In *MobiHoc '21: The Twenty-second International Symposium on Theory, Algorithmic Foundations, and Protocol Design for Mobile Networks and Mobile Computing, Shanghai, China, 26-29 July*, 2021.

Liaoyuan Zeng, Peiran Yu, and Ting Kei Pong. Analysis and algorithms for some compressed sensing models based on L1/L2 minimization. *SIAM J. Optim.*, 31(2):1576–1603, 2021.

Xinwei Zhang, Mingyi Hong, Sairaj V. Dhople, Wotao Yin, and Yang Liu. Fedpd: A federated learning framework with adaptivity to non-iid data. *IEEE Trans. Signal Process.*, 69:6055–6070, 2021.

Shenglong Zhou and Geoffrey Ye Li. Communication-efficient admm-based federated learning. *CoRR*, abs/2110.15318, 2021. URL https://arxiv.org/abs/2110.15318.

Shenglong Zhou and Geoffrey Ye Li. Federated learning via inexact ADMM. *CoRR*, abs/2204.10607, 2022.

H. Zou and T. Hastie. Regularization and variable selection via the elastic net. *J. R. Statist. Soc. B*, 67(2): 301–320, 2005.

## A   Supplement for Experiment

**The details of the training models.**   For all datasets, we apply neural networks with only Fully-connected (FC) layers as training models. The size of the models is shown as Table 3. Our code is available at https://anonymous.4open.science/r/FIAELT-8CC5/.

**Hyperparameter choosing.**   The learning rates are 0.012 for synthetic datasets, and 0.009 for FEMNIST. For FedPD, FedDR, and FedProx, we follow Tran-Dinh et al. (2021) to select the hyper-parameters, including $\mu$ for FedProx, $\eta$ for FedPD, and $\eta, \alpha$ for FedDR. As for FedMid Yuan et al. (2021) and FedDualAvg Yuan et al. (2021), we also select the hyper-parameters working best for plotting the performance and comparison.

**Additional Results with Different Learning Rates**   Figure 5 shows how different learning rates affect the performance of our FIAME on the FEMNIST dataset.

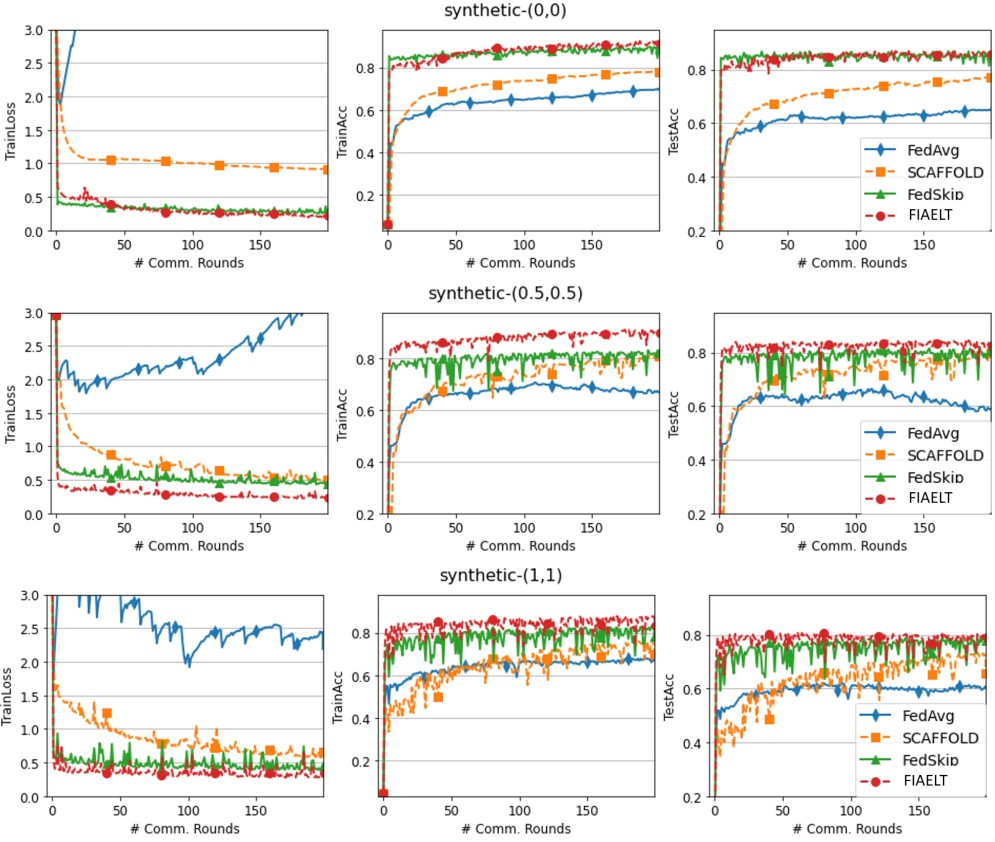

Figure 6: Results on Synthetic-{(0,0), (0.5, 0.5), (1,1)} dataset.

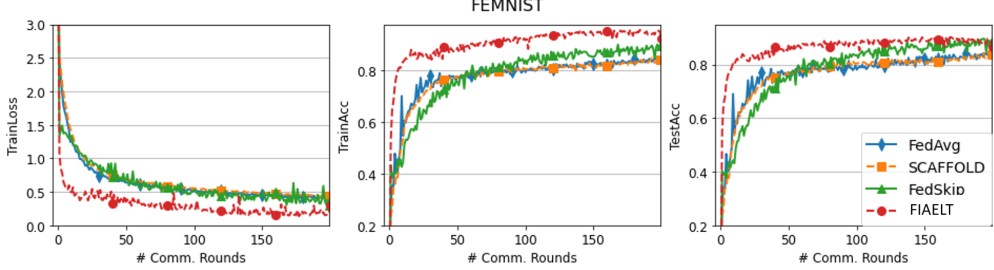

Figure 7: Results on FEMNIST dataset.

### A.1 Additional results comparing FIAME with non-ADMM based FL algorithms

We compare our method with FedAvg Li et al. (2020b), SCAFFOLD Karimireddy et al. (2020), FedSkip Fan et al. (2022).

**Results on synthetic datasets.** Following the data generation process on Li et al. (2020a); Tran-Dinh et al. (2021), we generate three datasets: synthetic-{(0,0), (0.5, 0.5), (1,1)}. All agents perform updates at each communication round. Our algorithm is compared using synthetic datasets in both iid and non-iid settings. The performance of 4 algorithms on non-iid synthetic datasets is shown as Figure 6. Our algorithm can achieve better results than FedAvg, SCAFFOLD, FedSkip on all three synthetic datasets.

**Results on FEMNIST dataset.** FEMNIST Cohen et al. (2017); Caldas et al. (2018) dataset is a more complex and federated extended MNIST. It has 62-class (26 upper-case and 26 lower-case letters, 10 digits) and the data is distributed to 200 devices. Figure 7 depicts the results of all 4 algorithms on FEMNIST. As it

shows, compared with the other 3 methods, FIAME has a significant improvement in both training accuracy and loss value. Our algorithm can also work much better with test accuracy than the other 3 algorithms.

## B  Convergence Analysis of Algorithm 1

**Proposition 1.** *If $X^* = (x_1^*, \ldots, x_p^*)$ is a stationary point of* (3)*, then $x_1^*$ is a stationary point of* (1)*. Furthermore, if $X = (x_1, \ldots, x_p)$ is an $\varepsilon$-stationary point of* (1)*, then $x_1$ is a $p\varepsilon$-stationary point of* (1)*.*

*Proof.* Note that

$$\mathfrak{C} = \left\{ (x_1, \ldots, x_p) : \ x_1 - x_2 = 0, \ x_2 - x_3 = 0, \ \ldots, \ x_{p-1} - x_p = 0 \right\}.$$

Using Theorem 6.14 of Rockafellar & Wets (1998), we have

$$N_\mathfrak{C} = \left\{ \sum_{i=1}^{p-1} \lambda_i (0, \ldots, 0, \underbrace{\mathbf{1}}_{\text{the } i_{\text{th}} \text{ coordinate}}, -\mathbf{1}, 0, \ldots, 0) : \ (\lambda_1, \ldots, \lambda_{p-1}) \in \mathbb{R}^{p-1} \right\},$$

where $\mathbf{1}$ is the vector in $\mathbb{R}^p$ whose coordinates are all one.

This together with Corollary 10.9, Proposition 10.5 shows that for any $Y \in \text{dom} \partial G$, $\partial G(Y)$ can be represetned as

$$\left\{ (\xi, 0, \ldots, 0) + \sum_{i=1}^{p-1} \lambda_i (0, \ldots, 0, \underbrace{\mathbf{1}}_{i_{\text{th}}}, -\mathbf{1}, 0, \ldots, 0) : \ \xi \in \partial g(y_1), \ (\lambda_1, \ldots, \lambda_{p-1}) \in \mathbb{R}^{p-1} \right\}. \tag{17}$$

Suppose $Y^* = (y_1^*, \ldots, y_p^*)$ is a stationary point of (3). Then $Y^* \in \text{dom} \partial G \subseteq \text{dom} G$. Thus, $y_1^* = \cdots = y_p^*$. In addition, it holds that

$$\begin{aligned}
0 &\in \nabla F(Y^*) + \partial G(Y^*) \\
&= (\nabla f_1(y^*), \ldots, \nabla f_p(y^*)) + (\partial g(y_1^*), 0, \ldots, 0) + \sum_{i=1}^{p} \lambda_i (0, \ldots, 0, \underbrace{\mathbf{1}}_{i_{\text{th}}}, -\mathbf{1}, 0, \ldots, 0),
\end{aligned} \tag{18}$$

where the second equality uses (17) together with Exercise 8.8 and Proposition 10.5 of Rockafellar & Wets (1998). The above relation is equivalent to

$$\begin{aligned}
0 &\in \nabla f_1(y^*) + \partial g(y_1^*) + \lambda_1 \mathbf{1} \\
0 &= \nabla f_2 - \lambda_1 \mathbf{1} + \lambda_2 \mathbf{1} \\
&\vdots \\
0 &= \nabla f_{p-1} - \lambda_{p-2} \mathbf{1} + \lambda_{p-1} \mathbf{1} \\
0 &= \nabla f_p(y^*) - \lambda_{p-1} \mathbf{1}.
\end{aligned} \tag{19}$$

Substituting $\lambda_1$ in (19) using the rest equality in the above relation, we have that

$$0 \in \sum_i \nabla f_i(y^*) + \partial g(y_1^*).$$

Thus $y^*$ is a stationary point of (1).

Now, suppose $Y = (y_1, \ldots, y_p)$ is a $\varepsilon$-stationary point of (3). Then $Y \in \text{dom} \partial G \subseteq \text{dom} G$. Thus, $y_1 = \cdots = y_p$ and

$$\varepsilon \geq d^2(0, \nabla F(Y) + \partial G(Y)). \tag{20}$$

Using (17) and Proposition 10.5 of Rockafellar & Wets (1998), we have that

$$
\begin{aligned}
&d^2(0, \nabla F(Y) + \partial G(Y)) \\
&= \min_{\xi \in \partial g(y_1), \lambda \in \mathbb{R}^{p-1}} \|\nabla f_1(y_1) + \xi + \lambda_1 \mathbf{1}\|^2 + \sum_{i=2}^{p-2} \|\nabla f_1(y_1) + \lambda_i \mathbf{1} - \lambda_{i-1} \mathbf{1}\|^2 \\
&\quad + \|\nabla f_p(y_1) - \lambda_{p-1} \mathbf{1}\|^2 \\
&\geq \min_{\xi \in \partial g(y_1), \lambda \in \mathbb{R}^{p-1}} \frac{1}{p} \| \sum_i \nabla f_i(y_1) + \xi\|^2\|^2 = \min_{\xi \in \partial g(y_1)} \frac{1}{p} \| \sum_i \nabla f_i(y_1) + \xi\|^2\|^2 \\
&= \frac{1}{p} d^2(0, \sum_i \nabla f_i(y_1) + \partial g(y_1)).
\end{aligned}
\tag{21}
$$

This together with (20) shows that $y_1$ is a $p\varepsilon$-stationary point. $\qquad\square$

### B.1 Proofs of Proposition 2

The problem in updating $Y^{t+1}$ in (6) is a constrained problem:

$$
\begin{aligned}
\min_Y \; & g(y_1) + \langle Z^t, X^{t+1} - Y \rangle + \frac{\beta}{2}\|X^{t+1} - Y\|^2 \\
\text{s.t. } & y_2 = y_3 = \cdots = y_p = y_1.
\end{aligned}
\tag{22}
$$

Since $\beta > L$, the objective in the above problem is strongly convex. Thus, there exists a unique solution $(y_1, y_2, \ldots, y_p)$ to (22). Denote the Lagrange multiplier for the above problem as $W = (w_1, w_2, \ldots, w_p)$. Then the Karush–Kuhn–Tucker condition for the above problem is

$$
0 \in \partial g(y_1) - z_1^{t+1} - \beta(x_1^{t+1} - y_1) - \sum_{i=2}^{p} w_i
\tag{23}
$$

$$
0 = -z_i^{t+1} + w_i - \beta(x_i^{t+1} - y_i), \; i = 2, \ldots, p
\tag{24}
$$

$$
y_i = y_1, \; i = 2, \ldots, p.
\tag{25}
$$

Combining (24) with (25) gives

$$
\sum_{i=2}^{p} w_i = \beta \sum_{i=2}^{p} (x_i^{t+1} - y_i) + \sum_{i=2}^{p} z_i^{t+1} = \beta \sum_{i=2}^{p} x_i^{t+1} - (p-1)\beta y_1 + \sum_{i=2}^{p} z_i^{t+1}.
$$

This together with (23) shows that

$$
\beta \sum_{i=2}^{p} x_i^{t+1} - (p-1)\beta y_1 + \sum_{i=2}^{p} z_i^{t+1} + z_1^{t+1} + \beta x_1^{t+1} \in \partial g(y_1) + \beta y_1,
$$

which is equivalent to

$$
\frac{1}{p} \sum_{i=1}^{p} (x_i^{t+1} + \frac{1}{\beta} z_i^{t+1}) \in \frac{1}{\beta p} \partial g(y_1) + y_1.
$$

This implies that $y_1 \in \mathrm{Prox}_{\frac{1}{\beta p} g}(\frac{1}{p} \sum_{i=1}^{p} (x_i^{t+1} + \frac{1}{\beta} z_i^{t+1}))$. Recalling (25), we deduce that the solution of the problem in the third equation of (6) is $(y_1, \ldots, y_1)$ with $y_1 = \mathrm{Prox}_{\frac{1}{\beta p} g}(\frac{1}{p} \sum_{i=1}^{p} (x_i^{t+1} + \frac{1}{\beta} z_i^{t+1})))$.

**Proposition 3.** *Consider* (1). *Set* $\beta > L := \max_i L_i$. *Let* $\{(x_i^t, y_i^t, z_i^t)\}$ *be generated by Algorithm 1. Using SVRG in Johnson & Zhang (2013) with Option II with frequency $m_i$, learning rate $\eta_i$, and initialization $x_i^t$ for* (8), *such that*

$$
\frac{1}{\eta_i(\beta - L_i)(1 - 2\eta_i(\beta + L_i))m_i} + \frac{2\eta_i(\beta + L_i)}{1 - 2\eta_i(\beta + L_i)} =: \rho_i < 1.
\tag{10}
$$

*Then criterion* (9) *is satisfied in at most* $k_t^i = \log_{1/\rho_i} \frac{\beta + L_i}{r_i(\beta - L_i)}$ *iterations of SVRG.*

*Proof.* Note that $L(x, y_i^t, z_i^t)$ is strongly convex with modulos $\beta - L_i$ and $\nabla L(x, y_i^t, z_i^t)$ is Lipschitz continuous with modulos $L_i + \beta$. Let $\rho_i := \frac{1}{(\beta - L_i)\eta(1 - 2\eta_i(\beta + L_i))m_i} + \frac{2\eta_i(\beta + L_i)}{1 - 2\eta_i(\beta + L_i)}$, where $m_i$ and $\eta_i$ is the frequency and learning rate in SVRG respectively. Using Theorem 1 of Johnson & Zhang (2013), there exists large $m$ such that

$$\mathbb{E}_i^t L_{\beta,i}(x_i^{t+1}, y^t, z_i^t) - L_{\beta,i}(x_{i,\star}^{t+1}, y^t, z_i^t) \leq \rho_i^{k_t}\left(L_{\beta,i}(x_i^t, y^t, z_i^t) - L_{\beta,i}(x_{i,\star}^{t+1}, y^t, z_i^t)\right) \tag{26}$$

Combing this with the strong convexity of $L(x, y_i^t, z_i^t)$ and the Lipschitz continuity of $\nabla L(x, y_i^t, z_i^t)$, we have that

$$\mathbb{E}_i^t \|x_i^{t+1} - x_{i,\star}^{t+1}\|^2 \leq \frac{\beta + L_i}{\beta - L_i}\rho_i^{k_t}\|x_i^t - x_{i,\star}^{t+1}\|^2 \leq r_i\|x_i^t - x_{i,\star}^{t+1}\|^2, \tag{27}$$

where the second inequality is based on $\frac{\beta + L_i}{\beta - L_i}\rho_i^{k_t} \leq r_i$. This completes the proof. $\qquad\square$

## C Proof for Convergence Analysis

To prove the results in Section Convergence Analysis of Algorithm 1, we first present the following well known facts for strongly convex functions, see Theorem 2 in Karimi et al. (2016) for example.

**Proposition 7.** *Let $f : \mathbb{R}^n \to \mathbb{R}$ be a strongly convex function with modulus $\mu$. Suppose in addition that $f$ is smooth and has Lipschitz continuous gradient with modulus $L$. Then there exists unique minimizer $x^*$ that minimize $f$ and it holds that*

$$\|\nabla f(x)\|^2 \geq 2\mu\left(f(x) - f(x^*)\right) \geq \mu^2\|x - x^*\|^2.$$

**Proposition 2.** *Consider* (3). *Let $\{(X^{t+1}, Y^{t+1}, Z^{t+1})\}$ be generated by* (6). *Suppose $\beta > \max_i L_i$. Then the solution of the problem in the third equation of* (6) *is $(y_1, \ldots, y_1)$ with $y_1 = \text{Prox}_{\frac{1}{\beta p}g}(\frac{1}{p}\sum_{i=1}^p (x_i^{t+1} + \frac{1}{\beta}z_i^{t+1})))$.*

The second and third relation in this proposition are obvious. We only need show that $X^t$ satisfies (11). Using (9) and the definition that $r = \max_i r$, we have

$$\mathbb{E}_i^t \|x_i^{t+1} - x_{i,\star}^{t+1}\|^2 \leq r_i\|x_i^t - x_{i,\star}^{t+1}\|^2 \leq r\|x_i^t - x_{i,\star}^{t+1}\|^2,$$

summing $i = 1, \ldots, p$, we obtain (11).

### C.1 Details and proofs of Proposition 5

Before proving Proposition 5, we first present several properties of the problem:

$$\min_X L_\beta(X, Y^t, Z^t), \tag{28}$$

where $Y^t$ and $z^t$ are defined as in Proposition 4.

**Proposition 8.** *Consider* (1). *Let $(X^t, Y^t, Z^t)$ be defined as in Proposition 4. Let $\beta \geq \sum_i L_i$. Denote $X_\star^{t+1} := \min_X L_\beta(X, Y^t, Z^{t+1})$.[1] Then the following statements hold:*

*(i) Denote $e^{t+1} = X^{t+1} - X_\star^{t+1}$. Then there exists $\xi^{t+1} \in \partial G(Y^{t+1})$ such that*

$$0 = \nabla F(X_\star^{t+1}) + Z^t + \beta(X_\star^{t+1} - Y^t) \Leftrightarrow -Z^t - \beta(X^{t+1} - e^{t+1} - Y^t) = \nabla F(X_\star^{t+1}) \tag{29}$$

*and*

$$0 = \xi^{t+1} - Z^{t+1} - \beta(X^{t+1} - Y^{t+1}) \tag{30}$$

*(ii) It holds that*

$$Z^{t+1} = (1 - \tau)Z^t + \beta\tau e^{t+1} + \tau\nabla F(X_\star^{t+1}) \tag{31}$$

---

[1]The existence and uniqueness of $X_\star^{t+1}$ are thanks to $\beta \geq \max_i L_i$ and Proposition 7.

*(iii) Let $r = \max_i r_i$. It holds that*

$$\mathbb{E}\|e^t\|^2 \leq \frac{2r}{1-2r}\mathbb{E}\|x^t - x^{t-1}\|^2 \tag{32}$$

*Proof.* (i) follows from the first optimality condition of (28) and (13). Combining (29) with (12), we have that

$$-Z^t - \frac{1}{\tau}(Z^{t+1} - Z^t) + \beta e^{t+1} = -Z^t - \beta(X^{t+1} - e^{t+1} - Y^t) = \nabla F(X^{t+1}_\star).$$
$$\Leftrightarrow Z^{t+1} = (1-\tau)Z^t + \beta\tau e^{t+1} + \tau\nabla F(X^{t+1}_\star).$$

Now, we bound $\mathbb{E}\|e^t\|^2$. Denote $e_i^t := x_i^t - x_{i*}^t$. Then using (27), we have that

$$\mathbb{E}_{t-1}\|e_i^t\|^2 \leq r_i\|x_i^{t-1} - x_{i*}^t\|^2 \leq 2r_i(\|x_i^t - x_i^{t-1}\|^2 + \|e_i^t\|^2).$$

where $c_i' := \frac{\beta + L_i}{\beta - L_i}$. Denote $c' = \max_i c_i'$, $\rho := \max_i \rho_i$ and $k_t := \max_i k_t^i$, $r = \max_i r_i$. Summing both sides of the above inequality from $i = 1, \ldots, p$, we obtain that

$$\mathbb{E}_{t-1}\|e^t\|^2 \leq 2r(\|x^t - x^{t-1}\|^2 + \mathbb{E}_{t-1}\|e^t\|^2).$$

Taking expectation on both sides over all randomness and rearranging the above inequality we obtain (32). $\qquad\square$

Now, we are ready to prove Proposition 5.

**Proposition 5.** *Select hyperparameters $\beta \geq 5L$, $r_i \in (0, 0.01]$, $\tau \in [1/2, 1)$. Denote $\Gamma := \frac{1-\tau}{\tau}$, $\Theta = 2\beta^2 + 4L^2$, $\Lambda := 4L^2$. $\Upsilon := \frac{\Theta}{\tau\beta}\frac{4r}{1-2r}$ and $\delta := \frac{1}{4}(\beta - L) - 2\Upsilon$. Define*

$$H(X, Y, Z, X', Z') := L_\beta(X, Y, Z) + \frac{\Gamma}{\tau\beta}\|Z - Z'\|^2 + \Upsilon\|X - X'\|^2.$$

*and $H_{t+1} := \mathbb{E}H(X^{t+1}, Y^{t+1}, Z^{t+1}, X^t, Z^t)$. Then for $t \geq 1$, it holds that $\delta \geq 0.1L$ and*

$$H_{t+1} \leq H_t - \delta\mathbb{E}\|X^{t+1} - X^t\|^2 - \frac{\beta}{2}\mathbb{E}\|Y^{t+1} - Y^t\|^2. \tag{14}$$

*Hence, the sequence $\{H_t\}$ converges to some $H_* \geq W$.*

*Proof.* Note that

$$\mathbb{E}_t L_\beta(X^{t+1}, Y^t, Z^t) - L_\beta(X^t, Y^t, Z^t) = L_\beta(X^{t+1}, Y^t, Z^t) - L_\beta(X_\star^{t+1}, Y^t, Z^t) + L_\beta(X_\star^{t+1}, Y^t, Z^t) - L_\beta(X^t, Y^t, Z^t)$$
$$\leq \rho^{k_t}\left(L_\beta(X^t, Y^t, Z^t) - L_\beta(X_\star^{t+1}, Y^t, Z^t)\right) + L_\beta(X_\star^{t+1}, Y^t, Z^t) - L_\beta(X^t, Y^t, Z^t)$$
$$\leq \rho^{k_t}\left(L_\beta(X^t, Y^t, Z^t) - L_\beta(X_\star^{t+1}, Y^t, Z^t)\right) - \frac{\beta - L}{2}\|X^t - X_\star^{t+1}\|^2$$
$$\leq \rho^{k_t}\left(L_\beta(X^t, Y^t, Z^t) - L_\beta(X_\star^{t+1}, Y^t, Z^t)\right) - \frac{\beta - L}{4}\mathbb{E}_t\|X^t - X^{t+1}\|^2 + \frac{\beta - L}{2}\mathbb{E}_t\|X^{t+1} - X_\star^{t+1}\|^2$$
$$\leq \rho^{k_t}\frac{\beta + L}{2}\|X^t - X_\star^{t+1}\|^2 - \frac{\beta - L}{4}\mathbb{E}_t\|X^t - X^{t+1}\|^2 + \frac{\beta - L}{2}\mathbb{E}_t\|e^{t+1}\|^2, \tag{33}$$

where the first inequality makes use of (26), the second inequality is because $L_\beta(X, Y^t, Z^t)$ is strongly convex with modulus $\beta - \max_i L_i$ and $X_\star^{t+1}$ is the minimizer of $\min_X L_\beta(X, Y^t, Z^t)$, the third inequality uses Young's inequality, the last inequality uses the Lipschitz continuity of $\nabla_X L_\beta(X, Y^t, Z^t)$.

Using the fact that $\|X^t - X_\star^{t+1}\|^2 \leq 2\mathbb{E}_t\|X^t - X^{t+1}\|^2 + 2\mathbb{E}_t\|e^{t+1}\|^2$, (33) can be further passed to

$$
\begin{aligned}
&\mathbb{E}_t L_\beta(X^{t+1}, Y^t, Z^t) - L_\beta(X^t, Y^t, Z^t) \\
&\leq 2\rho^{k_t}\frac{\beta+L}{2}\mathbb{E}_t\|X^t - X^{t+1}\|^2 + 2\rho^{k_t}\frac{\beta+L}{2}\mathbb{E}_t\|e^{t+1}\|^2 - \frac{\beta-L}{4}\mathbb{E}_t\|X^t - X^{t+1}\|^2 + \frac{\beta-L}{2}\mathbb{E}_t\|e^{t+1}\|^2 \\
&= \left(2\rho^{k_t}\frac{\beta+L}{2} - \frac{\beta-L}{4}\right)\mathbb{E}_t\|X^t - X^{t+1}\|^2 + \left(2\rho^{k_t}\frac{\beta+L}{2} + \frac{\beta-L}{2}\right)\mathbb{E}_t\|e^{t+1}\|^2 \\
&\leq \left(2\rho^{k_t}\frac{\beta+L}{2} - \frac{\beta-L}{4} + \left(2\rho^{k_t}\frac{\beta+L}{2} + \frac{\beta-L}{2}\right)\frac{2r}{1-2r}\right)\mathbb{E}_t\|X^t - X^{t+1}\|^2 \\
&= \left(\frac{\rho^{k_t}}{1-2r}(\beta+L) - \left(\frac{1}{4} - \frac{r}{1-2r}\right)(\beta-L)\right)\mathbb{E}_t\|X^t - X^{t+1}\|^2
\end{aligned}
\tag{34}
$$

where the second inequality uses (32).

Next, using (12), we have

$$
L_\beta(X^{t+1}, Y^t, Z^{t+1}) - L_\beta(X^{t+1}, Y^t, Z^t) = \frac{1}{\tau\beta}\|Z^{t+1} - Z^t\|^2 \tag{35}
$$

When $\tau \in (0, 1)$, combining (31) and the convexity of $\|\cdot\|^2$, we have that

$$
\begin{aligned}
\|Z^{t+1} - Z^t\|^2 &\leq (1-\tau)\|Z^t - Z^{t-1}\|^2 + \tau\|\beta(e^{t+1} - e^t) + \nabla(F(X_\star^{t+1}) - F(X_\star^t))\|^2 \\
&\leq (1-\tau)\|Z^t - Z^{t-1}\|^2 + 2\tau\beta^2\|e^{t+1} - e^t\|^2 + 2\tau\|\nabla(F(X_\star^{t+1}) - F(X_\star^t))\|^2 \\
&\leq (1-\tau)\|Z^t - Z^{t-1}\|^2 + 2\tau\beta^2\|e^{t+1} - e^t\|^2 + 2\tau L^2\|X_\star^{t+1} - X_\star^t\|^2,
\end{aligned}
$$

where the second inequality uses the Young's inequality for product, and the last inequality uses the Lipschitz continuity of $\nabla F$. Rearranging the above inequality, we have that

$$
\begin{aligned}
&\|Z^{t+1} - Z^t\|^2 \\
&\leq \frac{1-\tau}{\tau}\left(\|Z^t - Z^{t-1}\|^2 - \|Z^{t+1} - Z^t\|^2\right) + 2\beta^2\|e^{t+1} - e^t\|^2 + 2L^2\|X_\star^{t+1} - X_\star^t\|^2 \\
&\leq \frac{1-\tau}{\tau}\left(\|Z^t - Z^{t-1}\|^2 - \|Z^{t+1} - Z^t\|^2\right) + 2\beta^2\|e^{t+1} - e^t\|^2 \\
&\quad + 2L^2\left((1+\kappa^2)\|X^{t+1} - X^t\|^2 + (1+\kappa^{-2})\|e^{t+1} - e^t\|^2\right) \\
&= \frac{1-\tau}{\tau}\left(\|Z^t - Z^{t-1}\|^2 - \|Z^{t+1} - Z^t\|^2\right) + \left(2\beta^2 + 4L^2\right)\|e^{t+1} - e^t\|^2 \\
&\quad + 4L^2\|X^{t+1} - X^t\|^2,
\end{aligned}
\tag{36}
$$

where $\kappa > 0$ and the last inequality uses the definition of $e^{t+1}$ and Young's inequality for products.

Using the definition of $\Gamma$, $\Theta$ and $\Lambda$, (36) becomes

$$
\left\|Z^{t+1} - Z^t\right\|^2 \leq \Gamma\left(\|Z^{t-1} - Z^t\|^2 - \|Z^{t+1} - Z^t\|^2\right) + \Theta\|e^t - e^{t+1}\|^2 + \Lambda\left\|X^t - X^{t+1}\right\|^2. \tag{37}
$$

Now, combining (34), (35) and (37), we obtain that

$$
\begin{aligned}
&\mathbb{E}_t L_\beta(X^{t+1}, Y^t, Z^{t+1}) \\
&\leq L_\beta(X^t, Y^t, Z^t) + \left(\frac{\rho^{k_t}}{1-2r}(\beta+L) - \left(\frac{1}{4} - \frac{r}{1-2r}\right)(\beta-L)\right)\mathbb{E}_t\|X^t - X^{t+1}\|^2 \\
&\quad + \frac{\Gamma}{\tau\beta}\left(\|Z^{t-1} - Z^t\|^2 - \mathbb{E}_t\|Z^{t+1} - Z^t\|^2\right) + \frac{\Theta}{\tau\beta}\mathbb{E}_t\|e^t - e^{t+1}\|^2 + \frac{\Lambda}{\tau\beta}\mathbb{E}_t\left\|X^t - X^{t+1}\right\|^2 \\
&= L_\beta(X^t, Y^t, Z^t) + \left(\frac{\rho^{k_t}}{1-2r}(\beta+L) - \left(\frac{1}{4} - \frac{r}{1-2r}\right)(\beta-L)\right)\|X^t - X^{t+1}\|^2 \\
&\quad + \frac{\Gamma}{\tau\beta}\left(\|Z^{t-1} - Z^t\|^2 - \mathbb{E}_t\|Z^{t+1} - Z^t\|^2\right) + \frac{\Theta}{\tau\beta}\mathbb{E}_t\|e^t - e^{t+1}\|^2.
\end{aligned}
$$

Taking expectations with respect to $\mathcal{X}^t$, the above inequality implies

$$\mathbb{E}L_\beta(X^{t+1}, Y^t, Z^{t+1}) \leq \mathbb{E}L_\beta(X^t, Y^t, Z^t)$$
$$+ \left( \frac{\rho^{k_t}}{1-2r}(\beta+L) - \left(\frac{1}{4} - \frac{r}{1-2r}\right)(\beta - L) \right) \mathbb{E}\|X^t - X^{t+1}\|^2 \qquad (38)$$
$$+ \frac{\Gamma}{\tau\beta}\left(\mathbb{E}\|Z^{t-1} - Z^t\|^2 - \mathbb{E}\|Z^{t+1} - Z^t\|^2\right) + \frac{\Theta}{\tau\beta}\mathbb{E}\|e^t - e^{t+1}\|^2.$$

Combining (32) with (38), we obtain that

$$\mathbb{E}L_\beta(X^{t+1}, Y^t, Z^{t+1}) \leq \mathbb{E}L_\beta(X^t, Y^t, Z^t)$$
$$+ \left( \frac{\rho^{k_t}}{1-2r}(\beta+L) - \left(\frac{1}{4} - \frac{r}{1-2r}\right)(\beta - L) \right) \mathbb{E}\|X^t - X^{t+1}\|^2$$
$$+ \frac{\Gamma}{\tau\beta}\left(\mathbb{E}\|Z^{t-1} - Z^t\|^2 - \mathbb{E}\|Z^{t+1} - Z^t\|^2\right) \qquad (39)$$
$$+ \frac{\Theta}{\tau\beta}\frac{4r}{1-2r}\mathbb{E}\|X^t - X^{t-1}\|^2 + \frac{\Theta}{\tau\beta}\frac{4r}{1-2r}\mathbb{E}\|X^t - X^{t+1}\|^2.$$

Recall that $k_t = \min_i k_t^i$, $L = \max_i L_i$, $\rho = \max_i \rho_i$, $r = \max_i r_i$ and $k_t^i$ satisfies $\frac{\beta+L}{\beta-L}\rho_i^{k_t} \leq r_i$. This implies

$$\rho^{k_t} \leq \frac{\beta - L}{\beta + L}r.$$

This together with (39) shows that

$$\mathbb{E}L_\beta(X^{t+1}, Y^t, Z^{t+1}) \leq \mathbb{E}L_\beta(X^t, Y^t, Z^t) - \frac{1}{4}(\beta - L)\mathbb{E}\|X^t - X^{t+1}\|^2$$
$$+ \frac{\Gamma}{\tau\beta}\left(\mathbb{E}\|Z^{t-1} - Z^t\|^2 - \mathbb{E}\|Z^{t+1} - Z^t\|^2\right) + \underbrace{\frac{\Theta}{\tau\beta}\frac{4r}{1-2r}}_{\Upsilon}\mathbb{E}\|X^t - X^{t-1}\|^2 + \frac{\Theta}{\tau\beta}\frac{4r}{1-2r}\mathbb{E}\|X^t - X^{t+1}\|^2. \qquad (40)$$

Finally, using the definition of $\delta$ and $\Upsilon$, (40) further implies

$$\mathbb{E}L_\beta(X^{t+1}, Y^t, Z^{t+1})$$
$$\leq \mathbb{E}L_\beta(X^t, Y^t, Z^t) - \delta\mathbb{E}\|X^t - X^{t+1}\|^2$$
$$+ \frac{\Gamma}{\tau\beta}\left(\mathbb{E}\|Z^{t-1} - Z^t\|^2 - \mathbb{E}\|Z^{t+1} - Z^t\|^2\right) \qquad (41)$$
$$+ \Upsilon\left(\mathbb{E}\|X^t - X^{t-1}\|^2 - \mathbb{E}\|X^{t+1} - X^t\|^2\right).$$

Next, noting that $Y^{t+1}$ is the minimizer of (13) which is $\beta$-strongly convex, it holds that

$$\mathbb{E}L_\beta(X^{t+1}, Y^{t+1}, Z^{t+1}) \leq \mathbb{E}L_\beta(X^{t+1}, Y^t, Z^{t+1}) - \frac{\beta}{2}\mathbb{E}\|Y^{t+1} - Y^t\|^2. \qquad (42)$$

Summing (42) and (41), we have that

$$\mathbb{E}L_\beta(X^{t+1}, Y^{t+1}, Z^{t+1})$$
$$\leq \mathbb{E}L_\beta(X^t, Y^t, Z^t) - \delta\mathbb{E}\|X^t - X^{t+1}\|^2 + \frac{\Gamma}{\tau\beta}\left(\mathbb{E}\|Z^{t-1} - Z^t\|^2 - \mathbb{E}\|Z^{t+1} - Z^t\|^2\right)$$
$$+ \Upsilon\left(\mathbb{E}\|X^t - X^{t-1}\|^2 - \mathbb{E}\|X^{t+1} - X^t\|^2\right) - \frac{\beta}{2}\mathbb{E}\|Y^{t+1} - Y^t\|^2.$$

Rearranging the above inequality and recalling the definition of $H(X, Y, Z, X', Z')$, we have that

$$\mathbb{E}H(X^{t+1}, Y^{t+1}, Z^{t+1}, X^t, Z^t)$$
$$\leq \mathbb{E}H(X^t, Y^t, Z^t, X^{t-1}, Z^{t-1}) - \delta\mathbb{E}\|X^t - X^{t+1}\|^2 - \frac{\beta}{2}\mathbb{E}\|Y^{t+1} - Y^t\|^2.$$

Now we prove $\{H_t\}$ is convergent. Inequality (14) implies that $\{H_t\}$ is nonincreasing. Since $F$ and $G$ are bounded from below, we denote $W = \inf F + \inf G$. Now we show that $H_t \geq W$ for all $t$. Suppose to the contrary that there exists $t_0$ such that $H_{t_0} < W$. Since (14) implies $H_t$ is nonincreasing, it hold that

$$\sum_{t=1}^{T}(H_t - W) \leq \sum_{t=1}^{t_0-1}(H_t - W) + (T - t_0 + 1)(H_{t_0} - W).$$

Thus

$$\lim_{T\to\infty}\sum_{t=1}^{T}(H_t - W) = -\infty. \tag{43}$$

On the other hand, using (41), for $t \geq 1$, it holds that

$$
\begin{aligned}
H_t - W &\geq \mathbb{E}H(X^{t+1}, Y^{t+1}, Z^{t+1}, X^t, Z^t) - W \overset{(a)}{\geq} \mathbb{E}L_\beta(X^{t+1}, Y^t, Z^{t+1}) - W \\
&\geq \mathbb{E}F(X^{t+1}) + G(Y^t) + \langle X^{t+1} - Y^t, Z^{t+1}\rangle - W \\
&\geq \mathbb{E}\langle X^{t+1} - Y^t, Z^{t+1}\rangle \overset{(b)}{=} \frac{1}{\tau\beta}\mathbb{E}\langle Z^{t+1} - Z^t, Z^{t+1}\rangle = \frac{1}{\tau\beta}\left(\mathbb{E}\|Z^{t+1}\|^2 - \mathbb{E}\|Z^t\|^2 + \mathbb{E}\|Z^{t+1} - Z^t\|^2\right) \\
&\geq \frac{1}{\tau\beta}(\mathbb{E}\|Z^{t+1}\|^2 - \mathbb{E}\|Z^t\|^2).
\end{aligned}
$$

where (a) makes use of the definition of $H_t$ and $L_\beta$, (b) uses (12). Summing the above inequality from $t = 0$ to $T$ and take $T$ to the infinity, we have that

$$
\begin{aligned}
\lim_{T\to\infty}\sum_{t=1}^{T}(H_t - W) &\geq \lim_{T\to\infty}\sum_{t=1}^{T}\frac{1}{\tau\beta}(\|Z^{t+1}\|^2 - \|Z^t\|^2) \\
&= \frac{1}{\tau\beta}\lim_{T\to\infty}(\mathbb{E}\|Z^{T+1}\|^2 - \mathbb{E}\|Z^0\|^2) \geq -\frac{1}{\tau\beta}\|Z^0\|^2 > -\infty,
\end{aligned}
$$

which contradicts with (43). Therefore, $H_t$ is bounded from below. This together with (14) gives that $\{H_t\}$ is convergent. $\qquad\square$

### C.2 Details and proofs of Corollary 1

Thanks to Proposition 5, we have the following properties with respect to the successive changes.

**Corollary 4.** *Consider* (1) *and let* $(X^t, Y^t, Z^t)$ *be defined as in Proposition 4. Suppose assumptions in Proposition 5 hold. Then the following statements hold.*

*(i) It holds that*

$$\sum_{t=0}^{T}\mathbb{E}\|X^t - X^{t+1}\|^2 + \sum_{t=0}^{T}\mathbb{E}\|Y^{t+1} - Y^t\|^2 \leq \frac{L_\beta(X^0, Y^0, Z^0) + C - H_*}{\min\{\delta, \frac{\beta}{2}\}}. \tag{44}$$

*and*

$$
\begin{aligned}
\sum_{t=0}^{T}\mathbb{E}\|Z^t - Z^{t+1}\|^2 &\leq (1+\Gamma)\frac{3(r+1)}{(L-\beta)^2}\|\nabla L_\beta(X^0, Y^0, Z^0)\|^2 + 3\|X^0 - Y^0\|^2 \\
&\quad + 2\left(\Gamma + 2\Theta\frac{2r}{1-2r}\right)\frac{L_\beta(X^0, Y^0, Z^0) + C - W}{\min\{\delta, \frac{\beta}{2}\}},
\end{aligned} \tag{45}
$$

*where* $C := 2\tau\beta(\Gamma+1)\|X^0 - Y^0\|^2 + \frac{4}{(L-\beta)^2}\left(\frac{L+\beta+1}{2} + 2\tau\beta(\Gamma+1) + \Upsilon + \frac{(L-\beta)^2}{8}\right)\|\nabla_X L_\beta(X^0, Y^0, Z^0)\|^2.$ *with* $\Theta$ *and* $\Gamma$ *being defined as in Proposition 5.*

*(ii) It holds that*

$$\lim_t \mathbb{E}\|X^t - X^{t+1}\|^2 = \lim_t \mathbb{E}\|Y^{t+1} - Y^t\|^2 = \lim_t \mathbb{E}\|Z^{t+1} - Z^t\|^2 = \lim_t \mathbb{E}\|Y^t - X^t\|^2 = 0. \qquad (46)$$

*Proof.* Summing (14) from $t = 1$ to $T$, it holds that

$$
\begin{aligned}
H_T &\leq H_1 - \delta \sum_{t=1}^{T} \mathbb{E}\|X^t - X^{t+1}\|^2 - \frac{\beta}{2} \sum_{t=1}^{T} \mathbb{E}\|Y^{t+1} - Y^t\|^2 \\
&\leq H_1 - \delta \sum_{t=1}^{T-1} \mathbb{E}\|X^t - X^{t+1}\|^2 - \frac{\beta}{2} \sum_{t=1}^{T-1} \mathbb{E}\|Y^{t+1} - Y^t\|^2
\end{aligned}
\qquad (47)
$$

Now we bound $H_1$. Note that

$$
\begin{aligned}
H_1 &= \mathbb{E}L_\beta(X^1, Y^1, Z^1) + \frac{\Gamma}{\tau\beta} \mathbb{E}\|Z^1 - Z^0\|^2 + \Upsilon \mathbb{E}\|X^1 - X^0\|^2 \\
&\overset{(i)}{\leq} \mathbb{E}L_\beta(X^1, Y^0, Z^1) + \frac{\Gamma}{\tau\beta} \mathbb{E}\|Z^1 - Z^0\|^2 + \Upsilon \mathbb{E}\|X^1 - X^0\|^2 \\
&\overset{(ii)}{\leq} \mathbb{E}L_\beta(X^1, Y^0, Z^0) + \frac{\Gamma+1}{\tau\beta} \mathbb{E}\|Z^1 - Z^0\|^2 + \Upsilon \mathbb{E}\|X^1 - X^0\|^2 \\
&\overset{(iii)}{\leq} \mathbb{E}\Big( L_\beta(X^0, Y^0, Z^0) + \nabla_X L_\beta(X^0, Y^0, Z^0)^\top (X^1 - X^0) \\
&\quad + \frac{L+\beta}{2}\|X^1 - X^0\|^2 \Big) + \tau\beta(\Gamma+1)\mathbb{E}\|X^1 - Y^0\|^2 + \Upsilon \mathbb{E}\|X^1 - X^0\|^2 \\
&\leq L_\beta(X^0, Y^0, Z^0) + \frac{1}{2}\|\nabla_X L_\beta(X^0, Y^0, Z^0)\|^2 + 2\tau\beta(\Gamma+1)\|X^0 - Y^0\|^2 \\
&\quad + \Big( \frac{L+\beta+1}{2} + 2\tau\beta(\Gamma+1) + \Upsilon \Big) \mathbb{E}\|X^1 - X^0\|^2 \\
&\overset{(iv)}{\leq} L_\beta(X^0, Y^0, Z^0) + 2\tau\beta(\Gamma+1)\|X^0 - Y^0\|^2 \\
&\quad + \frac{4}{(L-\beta)^2} \Big( \frac{L+\beta+1}{2} + 2\tau\beta(\Gamma+1) + \Upsilon + \frac{(L-\beta)^2}{8} \Big) \|\nabla_X L_\beta(X^0, Y^0, Z^0)\|^2, \qquad (48)
\end{aligned}
$$

where (i) uses (42), (ii) uses (35), (iii) uses the property that $L_\beta(X, Y, \cdot)$ is $(L + \beta)$-smooth, and (iv) uses the following inequality.

$$
\begin{aligned}
\mathbb{E}\|X^1 - X^0\|^2 &\leq 2\mathbb{E}\|X^1 - X_*^1\|^2 + 2\mathbb{E}\|X^0 - X_*^1\|^2 \\
&\leq 4\mathbb{E}\|X^0 - X_*^1\|^2 \\
&\leq \frac{4}{(L-\beta)^2} \|\nabla_X L_\beta(X^0, Y^0, Z^0)\|^2
\end{aligned}
$$

Thus, summing (47) and (48), we have

$$
\begin{aligned}
H_T &\leq L_\beta(X^0, Y^0, Z^0) + C \\
&\quad - \delta \sum_{t=1}^{T} \mathbb{E}\|X^t - X^{t+1}\|^2 - \frac{\beta}{2} \sum_{t=1}^{T} \mathbb{E}\|Y^{t+1} - Y^t\|^2 \\
&\leq H_1 - \delta \sum_{t=1}^{T-1} \mathbb{E}\|X^t - X^{t+1}\|^2 - \frac{\beta}{2} \sum_{t=1}^{T-1} \mathbb{E}\|Y^{t+1} - Y^t\|^2.
\end{aligned}
$$

Rearranging the above inequality, we have that

$$
\begin{aligned}
\delta &\sum_{t=1}^{T-1} \mathbb{E}\|X^t - X^{t+1}\|^2 + \frac{\beta}{2} \sum_{t=1}^{T-1} \mathbb{E}\|Y^{t+1} - Y^t\|^2 \\
&\leq L_\beta(X^0, Y^0, Z^0) + C - H_T \leq L_\beta(X^0, Y^0, Z^0) + C - H_*,
\end{aligned}
\qquad (49)
$$

where the second inequality is because $\{H_t\}$ is nonincreasing and convergent. This implies (44).

Taking $T$ in the above inequality to infinity, we deduce that

$$\delta \sum_{t=0}^{\infty} \mathbb{E}\|X^t - X^{t+1}\|^2 + \frac{\beta}{2} \sum_{t=0}^{\infty} \mathbb{E}\|Y^{t+1} - Y^t\|^2 < \infty.$$

where the last inequality is because $\{H_t\}$ is convergent. Therefore, we have $\{\mathbb{E}\|X^t - X^{t+1}\|^2\}$, and $\lim_t \mathbb{E}\|Y^{t+1} - Y^t\|^2$ are summable and

$$\lim_t \mathbb{E}\|X^t - X^{t+1}\|^2 = \lim_t \mathbb{E}\|Y^{t+1} - Y^t\|^2 = 0. \tag{50}$$

In addition, summing (37) from $t = 1$ to $T$, we have that

$$\sum_{t=0}^{T} \mathbb{E}\|Z^t - Z^{t+1}\|^2$$

$$\leq (1+\Gamma)\|Z^0 - Z^1\|^2 + \Theta \sum_{t=1}^{T} \mathbb{E}\|e^t - e^{t+1}\|^2 + \Gamma \sum_{t=1}^{T} \mathbb{E}\|X^t - X^{t+1}\|^2$$

$$\leq (1+\Gamma)\|Z^0 - Z^1\|^2 + 2\Theta \frac{2r}{1-2r} \sum_{t=1}^{T} \mathbb{E}\|X^t - X^{t-1}\|^2 + (\Gamma + 2\Theta \frac{2r}{1-2r}) \sum_{t=0}^{T} \mathbb{E}\|X^t - X^{t+1}\|^2 \tag{51}$$

$$\leq (1+\Gamma)\|Z^0 - Z^1\|^2 + 2\left(\Gamma + 2\Theta \frac{2r}{1-2r}\right) \sum_{t=0}^{T} \mathbb{E}\|X^t - X^{t+1}\|^2$$

$$\leq (1+\Gamma)\|Z^0 - Z^1\|^2 + 2\left(\Gamma + 2\Theta \frac{2r}{1-2r}\right) \frac{L_\beta(X^0, Y^0, Z^0) + C - H_*}{\min\{\delta, \frac{\beta}{2}\}},$$

where the second inequality uses (32). Recall the definition of $Z^1$, we have that

$$\mathbb{E}\|Z^1 - Z^0\|^2 = \mathbb{E}\|X^1 - Y^0\|^2 \leq 3\mathbb{E}\|X^1 - X_\star^1\|^2 + 3\|X_\star^1 - X^0\|^2 + 3\|X^0 - Y^0\|^2$$

$$\leq 3r\|X^0 - X_\star^1\|^2 + 3\|X_\star^1 - X^0\|^2 + 3\|X_\star^1 - Y^0\|^2$$

$$\leq \frac{3(r+1)}{(L-\beta)^2}\|\nabla L_\beta(X^0, Y^0, Z^0)\|^2 + 3\|X^0 - Y^0\|^2.$$

This together with (51) gives

$$\sum_{t=0}^{T} \mathbb{E}\|Z^t - Z^{t+1}\|^2 \leq (1+\Gamma)\frac{3(r+1)}{(L-\beta)^2}\|\nabla L_\beta(X^0, Y^0, Z^0)\|^2 + 3\|X^0 - Y^0\|^2$$

$$+ 2\left(\Gamma + 2\Theta \frac{2r}{1-2r}\right) \frac{L_\beta(X^0, Y^0, Z^0) + C - H_*}{\min\{\delta, \frac{\beta}{2}\}}. \tag{52}$$

Taking $T$ in the above inequality to infinity we deduce that $\{\mathbb{E}\|Z^t - Z^{t+1}\|^2\}$ is summable and using (12), we have that

$$\lim \mathbb{E}\|Y^t - X^{t+1}\|^2 = \lim_t \mathbb{E}\|Z^t - Z^{t+1}\|^2 = 0.$$

This together with (50) gives that

$$\lim \mathbb{E}\|Y^t - X^t\|^2 = 0.$$

$\square$

### C.3 Details and proofs of Theorem 1

Here, we prove Theorem 1.

**Theorem 3.** *Consider* (1). *Let* $\{(x_1^t, \ldots, x_p^t, y^t, z_1^t, \ldots, z_p^t\}$ *be generated by Algorithm 1. Let* $(X^t, Y^t, Z^t)$ *be defined as in Proposition 4. Suppose assumptions in Proposition 5 hold. Then the following statements hold.*

(i) *There exists* $E > 0$ *such that*

$$\|\nabla F(Y^{t+1}) + \xi^{t+1}\| \leq E\left(\|X^{t+1} - X^t\| + \|Z^{t+1} - Z^t\| + \|Y^t - Y^{t+1}\|\right). \tag{53}$$

*where* $\xi^{t+1} \in \partial F(Y^{t+1})$.

(ii) *It holds that*

$$\frac{1}{1+T}\sum_{t=0}^{T}\mathbb{E}d^2(0, \nabla F(Y^{t+1}) + \partial G(Y^{t+1}))$$

$$\leq \frac{1}{T+1}R\left((1+\Gamma)\frac{3(r+1)}{(L-\beta)^2}\|\nabla L_\beta(X^0, Y^0, Z^0)\|^2 + 3\|X^0 - Y^0\|^2\right)$$

$$+ \frac{1}{T+1}R\left(2\Gamma + \Theta\frac{8r}{1-2r} + 2\right)\frac{L_\beta(X^0, Y^0, Z^0) + C - H_*}{\min\{\delta, \frac{\beta}{2}\}},$$

*where* $\Gamma$ *and* $\Theta$ *are defined in Proposition 5,* $H_*$ *and* $C$ *is defined in Proposition 5 and Corollary 4 respectively,* $R := \max\{3(L+\beta)^2\frac{2r}{1-2r}, \left(\frac{L}{\tau\beta}+1\right)^2, (L+\beta)^2\}$.

*Proof.* Using (29), it hold that

$$0 = \nabla F(Y^{t+1}) + \nabla F(X_\star^{t+1}) - \nabla F(Y^{t+1}) + Z^t + \beta(X_\star^{t+1} - Y^t).$$

Summing this with (30), we have that

$$0 = \nabla F(Y^{t+1}) + \xi^{t+1} + \nabla F(X_\star^{t+1}) - \nabla F(Y^{t+1}) + Z^t - Z^{t+1} + \beta(X_\star^{t+1} - X^{t+1}) - \beta(Y^{t+1} - Y^t).$$

This implies that

$$\|\nabla F(Y^{t+1}) + \xi^{t+1}\|$$
$$\leq \|\nabla F(X_\star^{t+1}) - \nabla F(Y^{t+1})\| + \|Z^t - Z^{t+1}\| + \beta\|X_\star^{t+1} - X^{t+1}\| + \beta\|Y^{t+1} - Y^t\|$$
$$\leq L\|X_\star^{t+1} - Y^{t+1}\| + \|Z^t - Z^{t+1}\| + \beta\|X_\star^{t+1} - X^{t+1}\| + \beta\|Y^{t+1} - Y^t\|$$
$$\leq L\|X_\star^{t+1} - X^{t+1}\| + L\|X^{t+1} - Y^t\| + (L+\beta)\|Y^t - Y^{t+1}\| + \|Z^t - Z^{t+1}\| \tag{54}$$
$$+ \beta\|X_\star^{t+1} - X^{t+1}\|$$
$$= (L+\beta)\|X_\star^{t+1} - X^{t+1}\| + \left(\frac{L}{\tau\beta}+1\right)\|Z^{t+1} - Z^t\| + (L+\beta)\|Y^t - Y^{t+1}\|,$$

where the last equality uses (12). Using (32), we have that $\mathbb{E}\|X_\star^{t+1} - X^{t+1}\|^2 \leq \sqrt{\frac{2r}{1-2r}}\mathbb{E}\|X^{t+1} - X^t\|^2$. Using this, (54) can be further passed to

$$\mathbb{E}\|\nabla F(Y^{t+1}) + \xi^{t+1}\|^2 \leq (L+\beta)\sqrt{\frac{2r}{1-2r}}3\mathbb{E}\|X^{t+1} - X^t\|^2 + \left(\frac{L}{\tau\beta}+1\right)3\mathbb{E}\|Z^{t+1} - Z^t\|^2$$

$$+ (L+\beta)3\mathbb{E}\|Y^t - Y^{t+1}\|^2.$$

This together with Cauchy-Schwarz inequality, we have that

$$\mathbb{E}\|\nabla F(Y^{t+1}) + \xi^{t+1}\|^2 \leq 3(L+\beta)^2\frac{2r}{1-2r}\mathbb{E}\|X^{t+1} - X^t\|^2 + \left(\frac{L}{\tau\beta}+1\right)^2\mathbb{E}\|Z^{t+1} - Z^t\|^2 \tag{55}$$

$$+ (L+\beta)^2\mathbb{E}\|Y^t - Y^{t+1}\|^2.$$

This proves (53).

Summing the above inequality from $t = 0$ to $T$, it holds that

$$\sum_{t=0}^{T} \mathbb{E}\|\nabla F(Y^{t+1}) + \xi^{t+1}\|^2$$

$$\leq 3(L+\beta)^2 \frac{2r}{1-2r} \sum_{t=0}^{T} \mathbb{E}\|X^{t+1} - X^t\|^2 + \left(\frac{L}{\tau\beta} + 1\right)^2 \sum_{t=0}^{T} \mathbb{E}\|Z^{t+1} - Z^t\|^2$$

$$+ (L+\beta)^2 \sum_{t=0}^{T} \mathbb{E}\|Y^t - Y^{t+1}\|^2$$

$$\leq \max\{3(L+\beta)^2 \frac{2r}{1-2r}, \left(\frac{L}{\tau\beta} + 1\right)^2, (L+\beta)^2\}$$

$$\cdot \left(\sum_{t=0}^{T} \mathbb{E}\|X^{t+1} - X^t\|^2 + \|Y^t - Y^{t+1}\|^2 + \|Z^{t+1} - Z^t\|^2\right)$$

$$\leq \max\{3(L+\beta)^2 \frac{2r}{1-2r}, \left(\frac{L}{\tau\beta} + 1\right)^2, (L+\beta)^2\}$$

$$\cdot \left((1+\Gamma)\frac{3(r+1)}{(L-\beta)^2}\|\nabla L_\beta(X^0, Y^0, Z^0)\|^2 + 3\|X^0 - Y^0\|^2 + \left(2\Gamma + \Theta\frac{8r}{1-2r} + 2\right)\frac{L_\beta(X^0, Y^0, Z^0) + C - H_*}{\min\{\delta, \frac{\beta}{2}\}}\right),$$

where $C := 2\tau\beta(\Gamma + 1)\|X^0 - Y^0\|^2 + \frac{4}{(L-\beta)^2}\left(\frac{L+\beta+1}{2} + 2\tau\beta(\Gamma + 1) + \Upsilon + \frac{(L-\beta)^2}{8}\right) \cdot \|\nabla_X L_\beta(X^0, Y^0, Z^0)\|^2$, the last inequality uses (44) and (45). Dividing both sides with $T + 1$ and recalling $\xi^{t+1} \in \partial G(Y^{t+1})$, we have the conclusion. Grouping the constants of $\|X^0 - Y^0\|^2$, $\|\nabla_X L_\beta(X^0, Y^0, Z^0)\|^2$, $L_\beta(X^0, Y^0, Z^0)$, we have that

$$\sum_{t=0}^{T} \mathbb{E}\|\nabla F(Y^{t+1}) + \xi^{t+1}\|^2 \tag{56}$$
$$\leq D\left(\|\nabla L_\beta(X^0, Y^0, Z^0)\|^2 + \|X^0 - Y^0\|^2 + L_\beta(X^0, Y^0, Z^0) - W\right),$$

where

$$D := \max\{3(L+\beta)^2 \frac{2r}{1-2r}, \left(\frac{L}{\tau\beta} + 1\right)^2, (L+\beta)^2\} \cdot \max\{D_1, D_2, D_3\} \tag{57}$$

with $D_1 := \frac{2\Gamma + \Theta\frac{8r}{1-2r} + 2}{\min\{\delta, \frac{1}{2}\beta\}}$, $D_2 := (1 + \Gamma)\frac{3(r+1)}{(L-\beta)^2} + D_1 \frac{4}{(L-\beta)^2}\left(\frac{L+\beta+1}{2} + 2\tau\beta(\Gamma + 1) + \Upsilon + \frac{(L-\beta)^2}{8}\right)$, $D_3 := \max\{3, D_1 2\tau\beta(\Gamma + 1)\}$.

$\square$

### C.3.1 Proofs of Proposition 6 and Corollary 3

We provide the detailed version of Proposition 6 as follows.

**Proposition 9.** *Consider* (1). *Let* $\{(x_1^t, \ldots, x_p^t, y^t, z_1^t, \ldots, z_p^t\}$ *be generated by Algorithm 1. Let* $(X^t, Y^t, Z^t)$ *be defined as in Proposition 4. Suppose assumptions in Proposition 5 hold. Suppose* $\{(X^t, Y^t, Z^t)\}$ *is bounded and denote the set of accumulation points of* $\{(X^t, Y^t, Z^t, X^{t-1}, Z^{t-1})\}$ *as* $\Omega$. *The following statements hold:*

*(i)* $\lim_t d((X^t, Y^t, Z^t, X^{t-1}, Z^{t-1}), \Omega) = 0$.

*(ii) Any accumulation point of* $\{Y^t\}$ *is a stationary point of* (1).

*(iii)* $H \equiv H_*$ *on* $\Omega$.

*Proof.* For (i), let $Y^*$ be an accumulation point of $\{Y^t\}$ with $Y^{t_i} \to Y^*$. Using (29) and (30), there exists $\xi^{t_i} \in G(Y^{t_i})$ such that

$$
\begin{aligned}
0 &= \nabla F(X_\star^{t_i}) + Z^{t_i-1} + \beta(X_\star^{t_i} - Y^{t_i-1}) \\
&= \nabla F(Y^t) + \nabla F(X_\star^{t_i}) - \nabla F(Y^t) + Z^{t_i-1} + \beta(X_\star^{t_i} - Y^{t_i-1}).
\end{aligned}
$$

and

$$
0 = \xi^{t_i} - Z^{t_i} - \beta(X^{t_i} - Y^{t_i}).
$$

The above relations shows that

$$
\begin{aligned}
0 &= \nabla F(Y^t) + \xi^{t_i} + \nabla F(X_\star^{t_i}) - \nabla F(Y^t) + Z^{t_i-1} - Z^{t_i} + \beta(X_\star^{t_i} - Y^{t_i-1}) - \beta(X^{t_i} - Y^{t_i}) \\
&= \nabla F(Y^t) + \xi^{t_i} + \nabla F(X_\star^{t_i}) - \nabla F(Y^t) + \tau\beta(X^{t_i} - Y^{t_i-1}) + \beta(X_\star^{t_i} - Y^{t_i-1}) - \beta(X^{t_i} - Y^{t_i})
\end{aligned}
\tag{58}
$$

where the equality makes uses of (12). Now we show that $\lim_i \|X_\star^t - X^t\| = 0$. Using Proposition 7 and (11), we have that

$$
\|e^t\|^2 = \|X_\star^t - X^t\|^2 \le \frac{2r}{1-2r}\|X^t - X^{t-1}\|^2.
$$

Since $\lim_t \|X^t - X^{t-1}\| = 0$, we have that

$$
\lim_i \|X_\star^t - X^t\| = 0. \tag{59}
$$

Next, we show that $\lim_i \|X^t - Y^{t-1}\| = 0$. Using (12), it holds that

$$
\begin{aligned}
&\|Z^t - Z^{t-1}\|^2 \\
&\le \Gamma\left(\|Z^{t-2} - Z^{t-1}\|^2 - \|Z^t - Z^{t-1}\|^2\right) + \Theta\|e^{t-1} - e^t\|^2 + \Lambda\|X^{t-1} - X^t\|^2 \\
&\le \Gamma\left(\|Z^{t-2} - Z^{t-1}\|^2 - \|Z^t - Z^{t-1}\|^2\right) + \Theta\frac{4r}{1-2r}\|X^{t-1} - X^{t-2}\|^2 + (\Lambda + \frac{4r}{1-2r})\|X^{t-1} - X^t\|^2
\end{aligned}
$$

where the first inequality uses (37) and the second inequality is due to (32). Summing the above inequality from $t=1$ to $T$, we have that

$$
\begin{aligned}
&\sum_{1=1}^T \|Z^t - Z^{t-1}\|^2 \le \Gamma\left(\|Z^{t_1-2} - Z^{t_1-1}\|^2 - \|Z^{t_K} - Z^{t_K-1}\|^2\right) \\
&\quad + \frac{1}{\tau\beta}\Theta\frac{4r}{1-2r}\sum_{1=1}^T \|X^{t-1} - X^{t-2}\|^2 + (\Lambda + \frac{4r}{1-2r})\sum_{1=1}^T \|X^{t-1} - X^t\|^2 \\
&\le \Gamma\left(\|Z^{t_1-2} - Z^{t_1-1}\|^2 - \|Z^{t_K} - Z^{t_K-1}\|^2\right) + \Theta\frac{4r}{1-2r}\sum_{i=1}^K \|X^{t-1} - X^{t-2}\|^2 \\
&\quad + (\Lambda + \frac{4r}{1-2r})\sum_{i=1}^K \|X^{t-1} - X^t\|^2 \\
&\le \Gamma\|Z^{t_1-2} - Z^{t_1-1}\|^2 + \Theta\frac{4r}{1-2r}\sum_{1=1}^T \|X^{t-1} - X^{t-2}\|^2 + (\Lambda + \frac{4r}{1-2r})\sum_{1=1}^T \|X^{t-1} - X^t\|^2.
\end{aligned}
$$

Taking $K$ in the above inequality to infinity and recalling that $\|X^{t-1} - X^t\|^2$ is summable, we deduce that $\sum_{1=1}^T \|Z^t - Z^{t-1}\|^2 < \infty$. This together with (12) show that

$$
\lim_t \|X^t - Y^{t-1}\| = \frac{1}{\tau\beta}\lim_t \|Z^t - Z^{t-1}\| = 0. \tag{60}
$$

Next, we show that $\lim_t \|Y^t - Y^{t-1}\| = 0$. Using (12) again, we have that

$$
Y^t - Y^{t-1} = X^{t+1} - X^t - \frac{1}{\tau\beta}(Z^{t+1} - Z^t) - \frac{1}{\tau\beta}(Z^t - Z^{t-1}).
$$

This together with the fact that $\lim_t \|X^t - X^{t-1}\| = \lim_t \|Z^t - Z^{t-1}\| = 0$ implies that $\lim_t \|Y^t - Y^{t-1}\| = 0$. Since $Y^{t_i} \to Y^*$, combining (59), (60) and (46), we have that

$$\lim_i Y^{t_i-1} = \lim_i X^{t_i} = \lim_i X_\star^{t_i} = \lim_i Y^{t_i} = Y^*.$$

This together with the continuity of $\nabla F$, the closedness of $\partial G$ and (58) shows that

$$0 \in \nabla F(Y^*) + \partial G(Y^*).$$

This completes the proof.

Now we prove (ii). Fix any $(X^*, Y^*, Z^*, \bar{X}^*, \bar{Z}^*) \in \Omega$. Then there exists $\{t_i\}_i$ such that $(X^{t_i}, Y^{t_i}, Z^{t_i}, X^{t_i-1}, Y^{t_i-1})$ converges to $(X^*, Y^*, Z^*, \bar{X}^*, \bar{Z}^*)$. Thanks to Proposition 5 (ii), we know that

$$H_* = \lim_i H(X^{t_i}, Y^{t_i}, Z^{t_i}, X^{t_i-1}, Y^{t_i-1}) \tag{61}$$

and

$$H(X^*, Y^*, Z^*, \bar{X}^*, \bar{Z}^*) = L_\beta(X^*, Y^*, Z^*) = F(X^*) + G(Y^*) + \langle X^* - Y^*, Z^* \rangle + \frac{\beta}{2}\|X^* - Y^*\|^2. \tag{62}$$

Since $Y^t$ is the minimizer of (13), it holds that

$$G(Y^{t_i}) + \langle X^{t_i} - Y^{t_i}, Z^{t_i} \rangle + \frac{\beta}{2}\|X^{t_i} - Y^{t_i}\|^2 \leq G(Y^*) + \langle X^{t_i} - Y^*, Z^{t_i} \rangle + \frac{\beta}{2}\|X^{t_i} - Y^*\|^2.$$

Taking the above inequality to infinity, we have that

$$\limsup_i G(Y^{t_i}) + \langle X^* - Y^*, Z^* \rangle + \frac{\beta}{2}\|X^* - Y^*\|^2$$

$$= \limsup_i G(Y^{t_i}) + \langle X^{t_i} - Y^{t_i}, Z^{t_i} \rangle + \frac{\beta}{2}\|X^{t_i} - Y^{t_i}\|^2$$

$$\leq G(Y^*) + \langle X^* - Y^*, Z^* \rangle + \frac{\beta}{2}\|X^* - Y^*\|^2.$$

This together with the closedness of $G$ shows that $\lim_i G(Y^{t_i}) = G(Y^*)$. This together with the continuity of $F$, Corollary 4 (ii) and (61) gives that

$$H_* = \lim_i H(X^{t_i}, Y^{t_i}, Z^{t_i}, X^{t_i-1}, Y^{t_i-1})$$

$$= F(X^*) + G(Y^*) + \langle X^* - Y^*, Z^* \rangle + \frac{\beta}{2}\|X^* - Y^*\|^2 = H(X^*, Y^*, Z^*, \bar{X}^*, \bar{Z}^*),$$

where the second equality uses (62). $\qquad\square$

**Corollary 3.** *Let $\{(x_1^t, \ldots, x_p^t, y^t, z_1^t, \ldots, z_p^t)\}$ be generated by Algorithm 1 with (9) holding deterministically. Let $(X^t, Y^t, Z^t)$ be defined as in Proposition 4. Suppose assumptions in Proposition 6 hold. Then any accumulation point of $\{y^t\}$ is a stationary point of (1).*

*Proof.* From Proposition 2, we understand that $Y^t = (y^t, \ldots, y_t)$ for any $t$. Let $y^*$ be any accumulation point of $y^t$. Then $Y^* = (y^*, \ldots, y^*)$ is an accumulation point of $\{Y^t\}$. Proposition 6 demonstrates that the $Y^*$ is a stationary point of (3). By applying Proposition 1, we deduce that $y^*$ is a stationary point of (1). $\quad\square$

### C.3.2  Details and proofs for Theorem 2

To show the global convergence of the generated sequence, we first need to bound the subdifferential of $\partial H(X^{t+1}, Y^{t+1}, Z^{t+1}, X^t, Z^t)$.

**Lemma 1.** *Consider* (1). *Let* $\{(x_1^t, \ldots, x_p^t, y^t, z_1^t, \ldots, z_p^t\}$ *be generated by Algorithm 1. Let* $(X^t, Y^t, Z^t)$ *be defined as in Proposition 4. Suppose* (9) *is satisfied deterministically (satisfied without expectation). Suppose assumptions in Proposition 5 hold. There exists* $D > 0$ *such that*

$$d(0, \partial H(X^{t+1}, Y^{t+1}, Z^{t+1}, X^t, Z^t)) \leq D \left( \|X^{t+1} - X^t\| + \|Y^{t+1} - Y^t\| + \|Z^{t+1} - Z^t\| \right).$$

*Proof.* Using Exercise 8.8, Proposition 10.5 and Corollary 10.9 of RockWets98, it holds that

$$\partial H(X, Y, Z, X', Z') \supseteq \begin{pmatrix} \nabla F(X) \\ \partial G(Y) \\ 0 \\ 0 \\ 0 \end{pmatrix} + \begin{pmatrix} Z + \beta(X - Y) + \frac{\Theta}{\tau\beta} \frac{16r}{1-2r}(X - X') \\ -Z - \beta(X - Y) \\ X - Y + \frac{2\Gamma}{\tau\beta}(Z - Z') \\ -\frac{\Theta}{\tau\beta} \frac{16r}{1-2r}(X - X') \\ -\frac{2\Gamma}{\tau\beta}(Z - Z'). \end{pmatrix}$$

Thus,

$$\begin{aligned} &\partial H(X^{t+1}, Y^{t+1}, Z^{t+1}, X^t, Z^t) \\ &\supseteq \begin{pmatrix} \nabla F(X^{t+1}) + Z^{t+1} + \beta(X^{t+1} - Y^{t+1}) + \frac{\Theta}{\tau\beta} \frac{16r}{1-2r}(X^{t+1} - X^t) \\ \partial G(Y^{t+1}) - Z^{t+1} - \beta(X^{t+1} - Y^{t+1}) \\ X^{t+1} - Y^{t+1} + \frac{2\Gamma}{\tau\beta}(Z^{t+1} - Z^t) \\ -\frac{\Theta}{\tau\beta} \frac{16r}{1-2r}(X^{t+1} - X^t) \\ -\frac{2\Gamma}{\tau\beta}(Z^{t+1} - Z^t) \end{pmatrix} \\ &\supseteq \begin{pmatrix} \nabla F(X^{t+1}) + Z^{t+1} + \beta(X^{t+1} - Y^{t+1}) + \frac{\Theta}{\tau\beta} \frac{16r}{1-2r}(X^{t+1} - X^t) \\ 0 \\ X^{t+1} - Y^{t+1} + \frac{2\Gamma}{\tau\beta}(Z^{t+1} - Z^t) \\ -\frac{\Theta}{\tau\beta} \frac{16r}{1-2r}(X^{t+1} - X^t) \\ -\frac{2\Gamma}{\tau\beta}(Z^{t+1} - Z^t) \end{pmatrix} \end{aligned} \tag{63}$$

where the seconde inclusion follows from (30).

Now, we bound each coordinate in the right hand side of the relation. For the first one, we denote $\mathcal{A}^{t+1} := \nabla F(X^{t+1}) + Z^{t+1} + \beta(X^{t+1} - Y^{t+1}) + \frac{\Theta}{\tau\beta} \frac{16r}{1-2r}(X^{t+1} - X^t)$. Using (29), we have that

$$\begin{aligned} \mathcal{A}^{t+1} \ni \ &\nabla F(X^{t+1}) - \nabla F(X_\star^{t+1}) + (Z^{t+1} - Z^t) \\ &+ \beta(X^{t+1} - Y^{t+1} - X_\star^{t+1} + Y^t) + \frac{\Theta}{\tau\beta} \frac{16r}{1-2r}(X^{t+1} - X^t). \end{aligned}$$

Thus, we deduce that $d^2(0, \mathcal{A}^{t+1})$ is bounded above by

$$\begin{aligned} &4(L + \beta)^2 \|X^{t+1} - X_\star^{t+1}\|^2 + 4\|Z^{t+1} - Z^t\|^2 + 4\beta^2 \|Y^t - Y^{t+1}\|^2 \\ &+ \frac{4\Theta^2}{\tau^2\beta^2} \frac{64r^2}{(1-2r)^2} \|X^{t+1} - X^t\|^2 \end{aligned} \tag{64}$$

where we also make use of the Lipscitz continuity of $\nabla F$.

For the third coordinate in (63), using (12), it holds that

$$\begin{aligned} \left\| X^{t+1} - Y^{t+1} + \frac{2\Gamma}{\tau\beta}(Z^{t+1} - Z^t) \right\|^2 &= \left\| \frac{1}{\tau\beta}(Z^{t+1} - Z^t) + Y^t - Y^{t+1} + \frac{2\Gamma}{\tau\beta}(Z^{t+1} - Z^t) \right\|^2 \\ &\leq 2\|Y^t - Y^{t+1}\|^2 + \frac{(1 + 2\Gamma)^2}{\tau^2\beta^2} \|Z^{t+1} - Z^t\|^2 \end{aligned}$$

This together with (63) and (64), we deduce that

$$
\begin{aligned}
& d^2(0, \partial H(X^{t+1}, Y^{t+1}, Z^{t+1}, X^t, Z^t)) \\
& \le 4(L+\beta)^2 \|X^{t+1} - X_\star^{t+1}\|^2 + 4\|Z^{t+1} - Z^t\|^2 + 4\beta^2\|Y^t - Y^{t+1}\|^2 \\
& + \frac{4\Theta^2}{\tau^2\beta^2} \frac{64 * 4r^2}{(1-2r)^2} \|X^{t+1} - X^t\|^2 + 2\|Y^t - Y^{t+1}\|^2 + \frac{(1+2\Gamma)^2}{\tau^2\beta^2}\|Z^{t+1} - Z^t)\|^2 \\
& + \frac{\Theta^2}{\tau^2\beta^2} \frac{64 * 4r^2}{(1-2r)^2} \|X^{t+1} - X^t\|^2 + \frac{4\Gamma^2}{\tau^2\beta^2}\|Z^{t+1} - Z^t\|^2.
\end{aligned}
\tag{65}
$$

Note that using 32, we have that

$$
\|X^{t+1} - X_\star^{t+1}\|^2 \le \frac{2r}{1-2r}\|X^{t+1} - X^t\|^2.
\tag{66}
$$

Combining (65) with (66), we have that

$$
\begin{aligned}
& d^2(0, \partial H(X^{t+1}, Y^{t+1}, Z^{t+1}, X^t, Z^t)) \\
& \le 4(L+\beta)^2 \frac{2r}{1-2r}\|X^{t+1} - X^t\|^2 + 4\|Z^{t+1} - Z^t\|^2 + 4\beta^2\|Y^t - Y^{t+1}\|^2 \\
& + \frac{4\Theta^2}{\tau^2\beta^2} \frac{64 * 4r^2}{(1-2r)^2} \|X^{t+1} - X^t\|^2 + 2\|Y^t - Y^{t+1}\|^2 + \frac{(1+2\Gamma)^2}{\tau^2\beta^2}\|Z^{t+1} - Z^t)\|^2 \\
& + \frac{\Theta^2}{\tau^2\beta^2} \frac{64 * 4r^2}{(1-2r)^2} \|X^{t+1} - X^t\|^2 + \frac{4\Gamma^2}{\tau^2\beta^2}\|Z^{t+1} - Z^t\|^2 \\
& = D'(\|X^{t+1} - X^t\|^2 + \|Y^t - Y^{t+1}\|^2 + \|Z^{t+1} - Z^t\|^2),
\end{aligned}
$$

where $D$ is the maximum of the coordinates of $\|X^{t+1} - X^t\|^2$, $\|Y^t - Y^{t+1}\|$ and $\|Z^{t+1} - Z^t\|^2$ on the right hand side of the above inequality. Finally, using the fact that $\sum_i^3 s_i^2 \le (\sum_i^3 a_i)^2$ for any $a_1, a_2, a_3 \ge 0$, the above inequality can be further passed to

$$
d^2(0, \partial H(X^{t+1}, Y^{t+1}, Z^{t+1}, X^t, Z^t)) \le D'(\|X^{t+1} - X^t\| + \|Y^t - Y^{t+1}\| + \|Z^{t+1} - Z^t\|).
$$

Taking square root on both sides of the above inequality we have the conclusion. $\qquad\square$

Now we are ready to prove Theorem 2. In fact, we already show the key properties that will be needed. They are Proposition 5, Corollary 4, Proposition C.3.1 and Lemma 1. The rest steps are routine. We follow the proofs in Borwein et al. (2017); Bolte et al. (2014); Li & Pong (2016) and include it only for completeness.

**Theorem 2.** *Consider* (1) *and Algorithm 1 with* (9) *holding deterministically. Let* $(X^t, Y^t, Z^t)$ *be defined as in Proposition 4. Suppose assumptions in Proposition 5 hold. Let $H$ be defined as in Proposition 5 and suppose $H$ is a KL function with exponent $\alpha \in [0, 1)$. Then $\{(X^t, Y^t, Z^t)\}$ converges globally. Denoting $(X^*, Y^*, Z^*) := \lim_t (X^t, Y^t, Z^t)$ and $d_s^t := \|(X^t, Y^t, Z^t) - (X^*, Y^*, Z^*)\|$, then the followings hold. If $\alpha = 0$, then $\{d_s^t\}$ converges finitely. If $\alpha \in (0, \frac{1}{2}]$, then there exist $b > 0$, $t_1 \in \mathbb{N}$ and $\rho_1 \in (0, 1)$ such that $d_s^t \le b\rho_1^t$ for $t \ge t_1$. If $\alpha \in (\frac{1}{2}, 1)$, then there exist $t_2$ and $c > 0$ such that $d_s^t \le ct^{-\frac{1}{4\alpha - 2}}$ for $t \ge t_2$.*

*Proof.* We first show that $\{(X^t, Y^t, Z^t)\}$ is convergent. If there exists $t_0$ such that $H_{t_0} = H_*$. Since $\{H_t\}$ is nonincreasing thanks to (14), we deduce that $H_t = H_*$ for all $t \ge t_0$. Using (14) again we have that for all $t \ge t_0$, it holds that $X^t = X^{t-1} = \cdots = X^{t_0-1}$ and $Y^t = Y^{t-1} = \cdots = Y^{t_0}$. Recalling in (46) we have that $\lim_t(X^t - Y^t) = 0$, we have that $Y^{t_0} = X^{t_0-1}$. Thus, $X^{t+1} - Y^t = Y^{t_0} - X^{t_0-1} = 0$ for all $t \ge t_0$. This together with (12), we deduce that $Z^{t+1} = Z^t = \cdots = Z^{t_0}$ for all $t \ge t_0$. Therefore, when there exists $t_0$ such that $H_{t_0} = H_*$, $\{(X^t, Y^t, Z^t)\}$ converge finitely.

Next, we consider the case where $H_t > H_*$ for all $t$. Thanks to Proposition C.3.1 (iii), using Lemma 6 of Bolte et al. (2014), there exists $r > 0$, $a > 0$ and $\psi \in \Psi_a$ such that

$$
\psi'(H(X, Y, Z, X'Z') - H_*)d(0, \partial H(X, Y, Z, X', Z')) \ge 1
$$

when $d((X, Y, Z, X', Z'), \Omega) \leq r$ and $H_* < H(X, Y, Z, X', Z') < H_* + a$. Thanks to Corollary 4 and Theorem 5, we know that there exists $t_1$ such that when $t > t_1$, $d((X^t, Y^t, Z^t, X^{t-1}, Z^{t-1}), \Omega) \leq r$ and $H_* < H(X^t, Y^t, Z^t, X^{t-1}, Z^{t-1}) < H_* + a$. Thus, it holds that

$$\psi'(H((X^t, Y^t, Z^t, X^{t-1}, Z^{t-1}) - H_*)d(0, \partial H((X^t, Y^t, Z^t, X^{t-1}, Z^{t-1})) \geq 1. \tag{67}$$

Recaling (14), we have that Since $\psi$ is concave, using the above inequality we have that

$$\begin{aligned}
\delta \|X^{t+1} - X^t\|^2 + \frac{\beta}{2}\|Y^{t+1} - Y^t\|^2 &\leq H_t - H_{t+1} \\
&\leq \psi'(H_t - H_*)d(0, \partial H(X^t, Y^t, Z^t, X^{t-1}, Z^{t-1}))(H_t - H_{t+1}) \\
&\leq \Delta_\psi^{t+1} d(0, \partial H(X^t, Y^t, Z^t, X^{t-1}, Z^{t-1}))
\end{aligned} \tag{68}$$

where the second inequality uses (67) and the last inequality uses the concavity of $\psi$. Using Lemma 1, we have from (68) that

$$\begin{aligned}
\frac{1}{2}\min\{\delta, \frac{\beta}{2}\}\left(\|X^{t+1} - X^t\| + \|Y^{t+1} - Y^t\|\right)^2 &\leq \min\{\delta, \frac{\beta}{2}\}\left(\|X^{t+1} - X^t\|^2 + \|Y^{t+1} - Y^t\|^2\right) \\
&\leq \delta\|X^{t+1} - X^t\|^2 + \frac{\beta}{2}\|Y^{t+1} - Y^t\|^2 \\
&\leq \Delta_\psi^{t+1} D\left(\|X^t - X^{t-1}\| + \|Y^t - Y^{t-1}\| + \|Z^t - Z^{t-1}\|\right)
\end{aligned} \tag{69}$$

where the first inequality uses the fact that $\frac{1}{2}(a+b)^2 \leq a^2 + b^2$ for any $a, b \in \mathbb{R}$.

Now we bound $\|Z^t - Z^{t-1}\|$. Using (31), we have that

$$\begin{aligned}
\|Z^{t+1} - Z^t\| &= |1 - \tau|\|Z^t - Z^{t-1}\| + \beta\tau\|e^{t+1} - e^t\| + \tau\|\nabla F(X_\star^{t+1}) - \nabla F(X_\star^t)\| \\
&\leq |1 - \tau|\|Z^t - Z^{t-1}\| + \beta\tau\|e^{t+1} - e^t\| + \tau L\|X_\star^{t+1} - X_\star^t\| \\
&\leq |1 - \tau|\|Z^t - Z^{t-1}\| + (\beta + L)\tau\|e^{t+1} - e^t\| + \tau L\|X^{t+1} - X^t\| \\
&\leq |1 - \tau|\|Z^t - Z^{t-1}\| + (\beta + L)\tau\frac{4}{(\beta - L)^2}\|X^t - X^{t-1}\| + \tau L\|X^{t+1} - X^t\|
\end{aligned}$$

where the second inequality uses the definition of $e^t$ and last inequality uses (32). Rearranging the above inequality, it holds that

$$\begin{aligned}
\|Z^t - Z^{t-1}\| &\leq \frac{1 + |1 - \tau|}{1 - |1 - \tau|}\left(\|Z^t - Z^{t-1}\| - \|Z^t - Z^{t+1}\|\right) - \|Z^t - Z^{t+1}\| \\
&+ \frac{2}{1 - |1 - \tau|}(\beta + L)\tau\frac{4}{(\beta - L)^2}\|X^t - X^{t-1}\| + \frac{2}{1 - |1 - \tau|}\tau L\|X^{t+1} - X^t\|.
\end{aligned}$$

Plugging this bound into (69), we have that

$$\begin{aligned}
\frac{1}{2}\min\{\delta, \frac{\beta}{2}\}&\left(\|X^{t+1} - X^t\| + \|Y^{t+1} - Y^t\|\right)^2 \\
&\leq \Delta_\psi^{t+1} D\left(\|X^t - X^{t-1}\| + \|Y^t - Y^{t-1}\|\right) \\
&+ \Delta_\psi^{t+1} D\left(\frac{1 + |1 - \tau|}{1 - |1 - \tau|}\left(\|Z^t - Z^{t-1}\| - \|Z^t - Z^{t+1}\|\right) - \|Z^t - Z^{t+1}\|\right) \\
&+ \Delta_\psi^{t+1} D\left(\frac{2(\beta + L)\tau}{1 - |1 - \tau|}\frac{4}{(\beta - L)^2}\|X^t - X^{t-1}\| + \frac{2\tau L}{1 - |1 - \tau|}\|X^{t+1} - X^t\|\right) \\
&\leq \Delta_\psi^{t+1} D D_1\left(\Delta_t^1 + \Delta_t^2\right),
\end{aligned}$$

where

$$\begin{aligned}
\Delta_\psi^{t+1} &:= \psi(H_t - H_*) - \psi(H_{t+1} - H_*), \\
D_1 &:= \max\{1 + \frac{2(\beta + L)\tau}{1 - |1 - \tau|}\frac{4}{(\beta - L)^2}, \frac{2\tau L}{1 - |1 - \tau|}, 1, \frac{1 + |1 - \tau|}{1 - |1 - \tau|}\}, \\
\Delta_t &:= \|X^t - X^{t-1}\| + \|X^{t+1} - X^t\| + \|Y^t - Y^{t-1}\|, \\
\Delta_t^2 &:= \left(\|Z^t - Z^{t-1}\| - \|Z^t - Z^{t+1}\|\right) - \|Z^t - Z^{t+1}\|.
\end{aligned}$$

Rearranging the above inequality and taking square toot on both sides, we obtain that

$$\|X^{t+1} - X^t\| + \|Y^{t+1} - Y^t\| \leq \sqrt{\frac{2}{\min\{\delta, \frac{\beta}{2}\}} \Delta_\psi^{t+1} DD_1 (\Delta_t^1 + \Delta_t^2)}$$

$$\leq \frac{2}{\min\{\delta, \frac{\beta}{2}\}} \Delta_\psi^{t+1} DD_1 + \frac{1}{4} (\Delta_t^1 + \Delta_t^2)$$

where the second inequality uses the fact that $\sqrt{ab} \leq \frac{1}{2}(a+b)$ for any $a, b > 0$. Recalling the definitions of $\Delta_t^1$ and $\Delta_t^2$, and rearranging the above inequality, we have that

$$\|X^{t+1} - X^t\| + \|Y^{t+1} - Y^t\| \leq \sqrt{\frac{2}{\min\{\delta, \frac{\beta}{2}\}} \Delta_\psi^{t+1} DD_1 \Delta}$$

$$\leq \frac{2}{\min\{\delta, \frac{\beta}{2}\}} \Delta_\psi^{t+1} DD_1$$

$$+ \frac{1}{4} \left( \|X^t - X^{t-1}\| + \|X^{t+1} - X^t\| + \|Y^t - Y^{t-1}\| \right)$$

$$+ \frac{1}{4} \left( \|Z^t - Z^{t-1}\| - \|Z^t - Z^{t+1}\| - \|Z^t - Z^{t+1}\| \right)$$

Further rearranging the above inequality, we have

$$\frac{1}{4}\|X^{t+1} - X^t\| + \frac{3}{4}\|Y^{t+1} - Y^t\| + \frac{1}{4}\|Z^t - Z^{t+1}\|$$

$$\leq \frac{2}{\min\{\delta, \frac{\beta}{2}\}} \Delta_\psi^{t+1} DD_1$$

$$+ \frac{1}{4} \left( \|X^t - X^{t-1}\| - \|X^{t+1} - X^t\| + \|Y^t - Y^{t-1}\| - \|Y^t - Y^{t+1}\| \right) \tag{70}$$

$$+ \frac{1}{4} \left( \|Z^t - Z^{t-1}\| - \|Z^t - Z^{t+1}\| \right).$$

Then, denoting $\Delta_{t+1} := \|X^{t+1} - X^t\| + \|Y^{t+1} - Y^t\| + D_2\|Z^{t+1} - Z^t\|$ (70) can be further passed to

$$\frac{1}{4}\Delta_{t+1} \leq \frac{2}{\min\{\delta, \frac{\beta}{2}\}} \Delta_\psi^{t+1} DD_1 + \frac{1}{4} (\Delta_t - \Delta_{t+1}) \tag{71}$$

Summing the above inequality from $t = t_1 + 1$ to $T$, we have that

$$\frac{1}{4} \sum_{t=t_1+1}^{T} \Delta_{t+1} \leq \frac{2}{\min\{\delta, \frac{\beta}{2}\}} \Delta_\psi^{t+1} DD_1 + \frac{1}{4} (\Delta_{t_1+1} - \Delta_{T+1})$$

$$\leq \frac{2}{\min\{\delta, \frac{\beta}{2}\}} \psi(H_t - H_*) DD_1 + \frac{1}{4}\Delta_{t_1+1}$$

where the last inequality uses the fact that $\psi > 0$. Taking $T$ in the above inequality to infinity, we see that $\sum_{t=t_1+1}^{\infty} \Delta_{t+1} < \infty$. Thus $\{(X^t, Y^t, Z^t)\}$ is convergent.

Next, we show the convergence rate of the generated sequence. Denote the limit of $(X^t, Y^t, Z^t)$ as $(X^*, Y^*, Z^*)$. Define $S_t = \sum_{i=t+1}^{\infty} \Delta_i$. Noting that $\|X^* - X^t\| + \|Y^* - Y^t\| + \|Z^t - Z^*\| \leq \sum_{i=t}^{\infty} \Delta_i = S_t$, it suffices to show the convergence rate of $S_t$. Using (71), there exists $D_2 > 0$ such that

$$S_t = \sum_{i=t}^{\infty} \Delta_i \leq D_2 \left( \psi(H_t - H_*) - \psi(H_{t+1} - H_*) \right) + (\Delta_t - \Delta_{t+1}) \tag{72}$$

$$\leq D_2\psi(H_t - H_*) + \Delta_t = D_2\psi(H_t - H_*) + (S_{t-1} - S_t).$$

Now we bound $\psi(H_t - H_*)$. From the KL assumption, $\psi(w) = cw^{1-\theta}$ with some $c >$. Thanks to Theorem 5 (ii) and (14), we have from the KL inequality, it holds that

$$c(1-\theta)d(0, \partial H(X^t, Y^t, Z^t, X^{t-1}, Z^{t-1})) \geq (H_t - H_*)^\theta. \tag{73}$$

Combining this with (1), we have that

$$c(1-\theta)D(S_{t-1} - S_t) \geq (H_t - H_*)^\theta.$$

This is equivalent to

$$c\left(c(1-\theta)D(S_{t-1} - S_t)\right)^{\frac{1-\theta}{\theta}} \geq c(H_t - H_*)^{1-\theta} = \psi(H_t - H_*).$$

Using this (72) can be further passed to

$$S_t \leq D_3(S_{t-1} - S_t)^{\frac{1-\theta}{\theta}} + (S_{t-1} - S_t), \tag{74}$$

where $D_3 := D_2 c \left(c(1-\theta)D\right)^{\frac{1-\theta}{\theta}}$. Now we claim

1. When $\theta = 0$, $\{(X^t, Y^t, Z^t)\}$ converges finitely.

2. When $\theta \in (0, \frac{1}{2}]$, there exist $a > 0$ and $\rho_1 \in (0,1)$ such that $S_t \leq a\rho_1^t$.

3. When $\theta \in (\frac{1}{2}, 1)$, there exists $D_4$ such that $S_t \leq ct^{-\frac{1-\theta}{2\theta-1}}$ for large $t$.

When $\theta = 0$, we claim that there exists $t$ such that $H_t = H_*$. Suppose to the contrary that $H_t > H_*$ for all $t$. Then, for large $t$, (73) holds, i.e., $d(0, \partial H(X^t, Y^t, Z^t, X^{t-1}, Z^{t-1})) \geq \frac{1}{c(1-\theta)} > 0$. However, thanks to 1 and Corollary 4, we know that $\lim_t d(0, \partial H(X^t, Y^t, Z^t, X^{t-1}, Z^{t-1})) = 0$, a contradiction. Therefore, there exists $t$ such that $H_t = H_*$. From the argument in the beginning of this proof, we see that $\{(X^t, Y^t, Z^t)\}$ converges finitely.

When $\theta \in (0, \frac{1}{2}]$, we have $\frac{1-\theta}{\theta} \geq 1$. Thanks to Corollary 4, we know that there exists $t_2$ such that $S_t - S_{t-1} < 1$. Thus, (74) can be further passed to $S_t \leq D_3(S_{t-1} - S_t) + (S_{t-1} - S_t)$. This implies that

$$S_t \leq \frac{D_3 + 1}{D_3 + 2} S_{t-1}.$$

Thus there exist $a > 0$ and $\rho_1 \in (0,1)$ such that $S_t \leq a\rho_1^t$.

When $\theta \in (\frac{1}{2}, 1)$, it holds that $\frac{1-\theta}{\theta} < 1$. From the last case, we know that $S_t - S_{t-1} < 1$ when $t > t_2$. Using (74), we have that $S_t \leq D_3(S_{t-1} - S_t)^{\frac{1-\theta}{\theta}} + (S_{t-1} - S_t)^{\frac{1-\theta}{\theta}} = (D_3 + 1)(S_{t-1} - S_t)^{\frac{1-\theta}{\theta}}$. This implies that

$$S_t^{\frac{\theta}{1-\theta}} \leq D_3^{\frac{\theta}{1+\theta}}(S_{t-1} - S_t).$$

With this inequality, following the arguments in Theorem 2 of Attouch & Bolte (2009) starting from Equation (13) in Attouch & Bolte (2009), there exists $c > 0$ such that $S_t \leq ct^{-\frac{1-\theta}{2\theta-1}}$ for large $t$. Thus, $\{S^t\}$ converges sublinearly. $\qquad\square$

