# OpenReview forum: "Inexact Alternating Direction Method of Multipliers with Efficient Local Termination Criterion for Cross-silo Federated Learning"
_TMLR — Withdrawn by Authors_

### Review · Reviewer_Q9LU · 2024-07-19

**Summary Of Contributions:**

The authors propose an inexact federated ADMM to solve the machine learning models with nonsmooth regularizer. They establish the $\mathcal{O}(1/T)$ for server updates and sequential convergence under K\L{} property. Experiments have also be conducted to demonstrate the efficiency of their algorithm.

**Audience:**

Yes

**Claims And Evidence:**

Yes

**Requested Changes:**

I noticed some typos. Please correct me if I am wrong.

* In (15), it seems you don not introduce the definition of $\xi^{t+1}$.

* In the line after equation (28), it should be capital $Z^t$.

* In (32), it should be capital $X^t-X^{t-1}$.

* In the line after equation (53), the $\xi^{t+1}\in \partial g(Y^{t+1})$.

**Strengths And Weaknesses:**

Strength:
* The paper is well-written.
* They establish the sequential convergence for federated learning algorithm.
* They also provide the local complexity for the local updates.

Weakness:
I have some questions and would appreciate your clarifications.

* From my understanding, your proposed stopping criteria still require the computation of the proximal operator, which may not be easily computed, especially in its exact form. In FedDR, they allow for an inexact computation of the proximal operator. I am curious whether your analysis allow such inexactness.

* I found the analysis of FedDR is also based on the sufficient decrease property. With a K\L{] assumption, I guess that they can also have the sequential convergence. Could you elaborate more on the advantages of your approach compared to FedDR?

* In FedDR, they also consider the asynchronous implementation. Will FIAELT work in asynchronous settings?

* In Theorem 2, you assume the K\L{} property of $H$. I think it would better to trace this back to the K\L{} property of $f_i$ and $g$.

---

### Review · Reviewer_pNCv · 2024-09-25

**Summary Of Contributions:**

This paper introduces a novel federated splitting method FIAELT for solving nonconvex and nonsmooth federated composite optimization in the context of cross-silo federated learning. The contributions include:
1. Propose a new criterion for the local update with $O(1)$ local complexity which can be quickly satisfied using SVRG. This new  criterion outperforms existing criterions with local updates unexplored or increases to infinity with T.
2. In the deterministic case, the authors prove that FIAELT has sequencial convergence properties; and under Kurdyka-Łojasiewicz (KL) geometry, FIAELT achieves global convergence.

**Audience:**

Yes

**Broader Impact Concerns:**

No broader impact is discussed.

**Claims And Evidence:**

Yes

**Requested Changes:**

Add analysis and experiments as above.

**Strengths And Weaknesses:**

Strengths:
1. FIAELT provides the first federated learning method that achieves sequential converegence when the KL property is satisfied by the function in nonconvex and nonsmooth case.
2. FIAELT provides a new local termination criterion which can be satisfied within iterations unrelated to $T$, which I believe would provide a new direction for researchers to design more efficient local update methods in federated learning.

Weaknesses:
1. As the sequential converegence are proved under the assumption that KL property is satisfied by the function, it would be better to provide some analyses for some cases when KL property is satisfied or not.
2. As the overall performance of those federated splitting methods depend both on local and server updates, it would be better to provide experimental results about the local complexities for satisfying the local update criterions and server complexities seperately comparing different federated splitting methods.

---

### Review · Reviewer_hyCQ · 2024-09-27

**Summary Of Contributions:**

This paper introduces the *Federated Inexact Alternating Direction Method of Multipliers with Efficient Local Termination (FIAELT)* for cross-silo federated learning. It proposes a new criterion for inexact $prox$ computation, using SVRG as a local solver that keeps the number of local updates depending only on the problem's parameters and does not increase with the number of iterations $T$. While doing this, the algorithm still maintains the state-of-the-art gradient convergence rate of $O(1/T)$ for server updates.

The authors also provide rigorous analysis proving sequential convergence to a stationary point and global convergence under the Kurdyka-Łojasiewicz (KL) geometry, with convergence rates ranging from sublinear to finite.

Finally, experiments demonstrate that FIAELT consistently outperforms competing methods in terms of training loss and accuracy, especially in scenarios involving nonsmooth regularizers.

**Audience:**

Yes

**Claims And Evidence:**

Yes

**Requested Changes:**

Questions:

1. Can this method be extended to the cross-device setup to account for the partial participation of clients?
2. Why was SVRG chosen as the local solver? Are there other algorithms that could achieve similar results?

**Strengths And Weaknesses:**

Strengths:

1. In real-world federated learning scenarios, devices often have limited computational resources, making it essential to minimize the number of local updates. This paper addresses this need by introducing a new criterion for approximate $prox$ solutions and employing SVRG to efficiently compute these solutions.

2. The method ensures that the total number of communication rounds preserves the state-of-the-art gradient convergence rate of $O(1/T)$, addressing one of the main bottlenecks in federated learning.

3. FIAELT exhibits sequential convergence properties. It proves that any accumulation point of the generated sequence at the server is a stationary point and achieves global convergence under the Kurdyka-Łojasiewicz (KL) geometry.

---

Weaknesses:

1. The method was compared to existing approaches in terms of communication rounds, which are consistent across the methods, as shown in Table 2. However, the primary distinction of this method is the number of local steps, and this is particularly important to emphasize in the experimental comparisons. I believe this should have been the main focus of the experiments.

2. Other ADMM-based federated learning methods account for cross-device federated learning by incorporating partial participation. This aspect was not addressed in this paper.

---

### Note · Authors · 2024-10-13

I have read and agree with the venue's withdrawal policy on behalf of myself and my co-authors.